# DELIFT: DATA EFFICIENT LANGUAGE MODEL INSTRUCTION FINE-TUNING

**Ishika Agarwal**[1], **Krishnateja Killamsetty**[2], **Lucian Popa**[2], **Marina Danilevsky**[2]
[1]University of Illinois Urbana-Champaign, [2]IBM Research
[1]ishikaa2@illinois.edu
[2]krishnateja.k@ibm.com, {lpopa, mdanile}@us.ibm.com

## ABSTRACT

Fine-tuning large language models (LLMs) is crucial for task specialization but often becomes resource-intensive due to redundant or uninformative data. Existing data selection methods typically rely either on computationally expensive gradient-based metrics or static embeddings that fail to adapt dynamically to the model's evolving state, thus limiting their practical effectiveness. To address this, we propose DELIFT (Data Efficient Language model Instruction Fine-Tuning), leveraging a novel, computationally efficient utility metric inspired by In-Context Learning (ICL). Our ICL-based metric measures the informational value of each data sample by quantifying its effectiveness as an in-context example in improving model predictions for other samples, reflecting its actual contribution relative to the model's current state. Integrated with tailored submodular optimization methods, DELIFT systematically selects diverse, informative subsets optimized specifically for each fine-tuning stage: instruction tuning, task-specific adaptation, and continual fine-tuning. Experimental results across multiple datasets and model scales show DELIFT reduces fine-tuning data requirements by up to 70% without compromising performance, consistently outperforming existing methods by up to 26% in effectiveness and efficiency.

## 1 INTRODUCTION

Large Language Models (LLMs) have become indispensable for solving a variety of natural language processing tasks, ranging from question answering and summarization to complex dialogue and reasoning (Brown et al., 2020; Touvron et al., 2023). Despite their remarkable adaptability, fine-tuning LLMs often requires enormous computational resources and time, especially when a significant portion of the training data is either redundant or uninformative (Gururangan et al., 2020; Sorscher et al., 2023). This challenge grows more critical with increasing model and dataset sizes, posing a key limitation to the broader deployment of LLMs.

Existing data selection methods generally fall under two paradigms: (1) *static embedding-based* approaches that compute sample similarities without reflecting the model's evolving state (Bukharin & Zhao, 2024; Chen et al., 2024), and (2) *gradient-based* methods that offer more model-specific feedback but often entail prohibitive computational overhead, especially for large-scale models (Killamsetty et al., 2021b; Xia et al., 2024). Although both paradigms can yield initial benefits, they often fail to account for how a model's knowledge shifts over multiple fine-tuning phases: **(1) Instruction Tuning** (Mishra et al., 2022; Wei et al., 2022; Longpre et al., 2023), which enhances the model's ability to follow diverse instructions; **(2) Task-Specific Fine-Tuning** (Gururangan et al., 2020; Cobbe et al., 2021), which focuses on refining domain expertise; and **(3) Continual Fine-Tuning** (Madotto et al., 2021; Wu et al., 2024), which incrementally incorporates new knowledge while mitigating catastrophic forgetting.

Thus, a natural question arises:

*Can we develop a unified, computationally efficient data selection framework that adapts to all stages of fine-tuning and maximizes model performance while minimizing data redundancy?*

| | Selected in the subset |
|---|---|
| | Pruned out of the subset |

**Use Case 1**: fine-tune a model to follow instructions. Subset should contain points that are **diverse**.

Dataset

| Instruction | Input | Output |
|---|---|---|
| Given the context, answer the question. | **Question**: Who is New Zealand's Prime Minister? **Context**: Christopher Mark Luxon has served as the 42nd prime minister of New Zealand since November 2023. | Christopher Mark Luxon |
| Given the context, answer the question. | **Question**: When did Luxon start his term? **Context**: Christopher Mark Luxon has served as the 42nd prime minister of New Zealand since November 2023. | November 2023 |
| Write a sentence with the given words. | Sun, park, dog. | Once the sun was up, I went to the park with my dog. |
| Classify the given objects into a category. | Crab, tuna, lobster. | Seafood. |

(a)

**Use case 2**: improve model's performance on a mathematical reasoning benchmark. Subset should contain points that are **diverse** *and* **representative of the benchmark.**

(Example) Benchmark Data

| Input | Output |
|---|---|
| Abby worked for 8 hours per day for 30 days. How much did she work? | 240 hours |
| Ben paid for his dinner ($20), Charles' dinner ($18) and Dennis' dinner ($15). How much did he pay? | $53 |
| Eunice has 20 oranges, and 4 friends. How many oranges does each friend get? | 5 oranges |
| Greg has 20 baseball cards and trades 5 of them. How many are left? | 15 cards |

Dataset

| Input | Output |
|---|---|
| Hannah had 40 nickels and won 10 more. How many nickels does she have? | 50 nickels |
| Fred had 25 roses and gave 10 to Mom. How many are left? | 15 roses |
| Lydia gave away ½ her pie to Mike and ¼ of her pie to Ned. How much of the pie is left? | ¼ of the pie |
| Is the following word positive or negative? "Happiness" | Positive |

(b)

**Use case 3**: continual learning on review sentiment analysis datasets. Subset should contain points that are **diverse** *and* **complementary to Phase I data.**

Previously Trained, Phase I Data

| Input | Output |
|---|---|
| This restaurant has good paella except that it is sometimes too salty. | Negative |
| The waiters are impatient and rude, they rushed me to order my food. | Negative |
| The atmosphere of this restaurant is cozy and comfortable, with dim lights. | Positive |
| The food came very quickly. | Positive |

New, Phase II Data

| Input | Output |
|---|---|
| The fried rice is amazing! | Positive |
| The camera resolution quality is low, and the lens do not focus properly. | Negative |
| This phone is lightweight, thin, and fits in my pockets easily. | Positive |
| The restaurant closes too early. | Negative |

(c)

Figure 1: DELIFT data selection across fine-tuning stages. (a) **Instruction Tuning**: Diverse instructions selected; redundant samples pruned. (b) **Task-Specific Fine-Tuning**: Mutually informative (with benchmark data) and diverse samples are prioritized for selection. (c) **Continual Fine-tuning**: New samples that are novel are integrated; new samples with overlapping information are pruned.

In this paper, we introduce DELIFT (Data-Efficient Language Model Instruction Fine-Tuning), a *single-stop solution* designed to address data selection across *all* fine-tuning stages within a single framework. DELIFT is *grounded in information theory* yet uses the practical intuition of in-context examples to assess the 'information gain' of each data sample relative to the current state of a model. Specifically, we propose a new utility metric that captures how effectively one sample improves the model's prediction of another. By combining these pairwise utilities with submodular optimization, DELIFT generates diverse, nonredundant subsets *uniquely tailored* to each fine-tuning phase.

We evaluated DELIFT on various tasks and model scales, consistently observing that it can prune up to 70% of the training data without hurting performance - and often improving it - outperforming existing methods by up to 26% in efficiency and effectiveness. In doing so, we show that *careful, utility-driven data selection can be far more effective* than sheer data volume, opening the door to more resource-friendly and targeted fine-tuning.

Our primary contributions are as follows.

**1. A unified information-theoretic data selection paradigm** that leverages pairwise utilities grounded in conditional pointwise mutual information, making it adaptable to instruction tuning, task-specific adaptation, and continual fine-tuning.

**2. A single-stop, submodular optimization framework** that integrates these utilities to provide diverse, high-value subsets for each fine-tuning stage without incurring prohibitive computation.

**3. Extensive empirical validation** showing up to 70% data reduction with minimal (and sometimes zero) performance loss across multiple domains, demonstrating substantial gains in both efficacy and efficiency.

The remainder of this paper is organized as follows. Section 2 reviews prior work on data-efficient strategies for fine-tuning LLMs and situates our approach within the literature. Section 3 introduces our information-theoretic utility metric and describes how it integrates with submodular optimization to enable data selection across diverse fine-tuning stages. Section 4 presents comprehensive experiments demonstrating the effectiveness and efficiency of our framework on multiple tasks and models. Finally, Section 5 discusses the broader implications of our results, outlines limitations, and suggests directions for future research. Our complete code base is publicly available at `https://github.com/agarwalishika/delift`, enabling further exploration and replication.

## 2 RELATED WORK

**Data Subset Selection for Deep Neural Networks.** Selecting an informative subset of training samples is a longstanding strategy to reduce computational costs and enhance model generalization. **Model-Independent Approaches.** Traditional *model-independent* techniques, such as clustering or distance metrics on pre-trained embeddings (Bukharin & Zhao, 2024; Du et al., 2023; Killamsetty et al., 2023), capture broad semantic similarities but do not reflect the model's changing state, limiting their effectiveness during iterative fine-tuning. **Model-Dependent Approaches.** *Model-dependent* methods incorporate the model's evolving knowledge by analyzing gradients or loss values (Killamsetty et al., 2021b;a; Xia et al., 2024), often outperforming static approaches. However, performing gradient or influence estimations at scale becomes prohibitively expensive for large models. Techniques like LESS (Xia et al., 2024) alleviate some overhead via parameter-efficient fine-tuning (e.g., LoRA), , yet still incur repeated gradient or influence calculations that scale poorly with dataset size. **Subset Selection with LLM Feedback.** Another emerging direction leverages LLM feedback to score or filter training samples. For instance, SelectIT (Liu et al., 2024) employs self-reflection prompts to rate data quality, while filtering approaches using GPT-4 (Chen et al., 2024) rely on external heuristics. Though these provide a form of model-aware sampling, they typically lack a principled theoretical grounding. *In addition, all these approaches primarily target a single fine-tuning stage, limiting their adaptability for instruction tuning, task-specific adaptation, or continual learning.*

**Our Contribution.** In contrast, we present a *unified*, information-theoretic framework that operates effectively across all fine-tuning stages: instruction tuning, task-specific adaptation, and continual fine-tuning. Our novel utility metric quantifies how one data point aids the prediction of another, mirroring the model's evolving knowledge. Integrated within a submodular selection paradigm (Fujishige, 2005; Iyer et al., 2021), this approach balances diversity, coverage, and informativeness throughout the entire fine-tuning pipeline. As a result, we bridge the gap left by existing methods that are either restricted to a single phase or computationally infeasible at scale, demonstrating consistent performance improvements and notable efficiency gains.

## 3 METHODOLOGY

Our goal is to efficiently identify a subset of data that maximizes the performance of large language models across three fine-tuning stages: **(1) Instruction Tuning**, **(2) Task-Specific Fine-Tuning**, and **(3) Continual Fine-Tuning**. This section first introduces our novel *information-theoretic* pairwise utility metric (Section 3.1) and then explains how we leverage submodular optimization to achieve data-efficient selection (Section 3.2). Finally, we show how these components unify into a *single-stop solution* for all fine-tuning stages (Section 3.3).

### 3.1 PAIRWISE UTILITY METRIC

Let $\mathcal{D} = \{(x_i, y_i)\}$ be a training set, where each $x_i$ is an input sequence and $y_i$ is the corresponding output. Consider two samples $(x_i, y_i)$ and $(x_j, y_j)$, and let $GT_i$ denote the ideal "ground truth" distribution that assigns probability 1 to the token sequence $y_i$ and 0 otherwise. We define $p(y_i \mid x_i)$ as the model's predicted probability distribution of $y_i$ given $x_i$ alone, and $p(y_i \mid x_i, x_j, y_j)$ as the predicted distribution of $y_i$ when $(x_j, y_j)$ is also provided (e.g., as an in-context example).

**Definition (Utility Function).** We capture the *information gain* of $(x_j, y_j)$ for predicting $(x_i, y_i)$ via:

$$UF_{ij} \;=\; d\big(GT_i,\, p(y_i \mid x_i)\big) \;-\; d\big(GT_i,\, p(y_i \mid x_i, x_j, y_j)\big), \tag{1}$$

where $d(\cdot, \cdot)$ is a distance between probability distributions.

**Information-Theoretic Interpretation.** Below we state a simplified version of our main theoretical result (see Appendix A for the full proof):

**Theorem 1** *(Informal Statement) If $d(\cdot, \cdot)$ is chosen to be the Kullback-Leibler (KL) divergence, then the utility $UF_{ij}$ coincides with the (conditional) pointwise mutual information between $y_i$ and $(x_j, y_j)$ given $x_i$. Formally,*

$$UF_{ij} \;=\; \log \frac{p(y_i \mid x_i, x_j, y_j)}{p(y_i \mid x_i)} \;=\; \sum_{t=1}^{T} \log\Big( \tfrac{p(y_{it} \mid x_i, x_j, y_j, y_{i,<t})}{p(y_{it} \mid x_i, y_{i,<t})} \Big).$$

Thus, $UF_{ij}$ captures how much $(x_j, y_j)$ *informs* the prediction of $y_i$ given $x_i$.

**Practical Computation.** In practice, *for numerical stability*, we adopt a length-normalized Euclidean distance rather than the KL-divergence:

$$d\big(GT_i,\, p(y_i \mid \cdot)\big) \;=\; \Big\| 1 - p(y_i \mid \cdot) \Big\|_2,$$

where we only extract the probability assigned to each ground-truth token under teacher forcing. This preserves the spirit of the PMI-based formulation while avoiding the instability issues often encountered with near-zero probabilities in large vocabularies. We compute $UF_{ij}$ for all pairs $(i, j)$ once before fine-tuning. Although this step is $O(n^2)$ in the dataset size, the cost is amortized because the *same utility matrix* can be reused for different fine-tuning stages or methods.

### 3.2 SUBMODULAR OPTIMIZATION FOR DATA SELECTION

After computing $UF_{ij}$, we define a kernel matrix $s_{ij}$ (e.g., set $s_{ij} = \max(UF_{ij}, 0)$) and utilize it in well-studied *submodular* functions (Fujishige, 2005; Iyer et al., 2021). Submodularity naturally captures diminishing returns and promotes coverage of diverse yet *informative* samples.

**Objectives.** We primarily adopt three variants:

1. **Facility Location (FL):** $f_{FL}(\mathcal{A}) = \sum_{i \in \mathcal{D}} \max_{j \in \mathcal{A}} s_{ij}$.

2. **Facility Location Mutual Information (FLMI):** $f_{FLMI}(\mathcal{A}; \mathcal{D}_T) = \sum_{i \in \mathcal{D}} \max_{j \in \mathcal{A}} s_{ij} + \eta \sum_{j \in \mathcal{A}} \max_{i \in \mathcal{D}_T} s_{ij}$.

3. **Facility Location Conditional Gain (FLCG):** $f_{FLCG}(\mathcal{A} \mid \mathcal{D}_E) = \sum_{i \in \mathcal{D}} \max\left(\max_{j \in \mathcal{A}} s_{ij} - \nu \max_{k \in \mathcal{D}_E} s_{ik}, 0\right).$

FL supports coverage of $\mathcal{D}$ itself, ideal for **instruction tuning**; FLMI additionally ties the selection to a **task-specific** dataset $\mathcal{D}_T$; and FLCG incorporates a previously used set $\mathcal{D}_E$ to promote **continual** learning of novel examples.

---

**Algorithm 1** Greedy Maximization for Submodular Function

---

**Require:** Dataset $\mathcal{D}$, submodular function $f$, budget $k$
1: Initialize subset $\mathcal{A} \leftarrow \emptyset$
2: **for** $t = 1$ to $k$ **do**
3:      Select $d^* = \arg\max_{d \in \mathcal{D} \setminus \mathcal{A}} (f(\mathcal{A} \cup \{d\}) - f(\mathcal{A}))$
4:      Update $\mathcal{A} \leftarrow \mathcal{A} \cup \{d^*\}$
5: **end for**
6: **return** $\mathcal{A}$

---

**Subset Selection.** To choose a subset $\mathcal{A}$ of size $k$, we apply a classic greedy heuristic (Nemhauser et al., 1978), each step picking $d^* = \arg\max_{d \in \mathcal{D} \setminus \mathcal{A}}[f(\mathcal{A} \cup \{d\}) - f(\mathcal{A})]$ (see Algorithm 1). This yields a $1 - \frac{1}{e}$ approximation factor for submodular functions, ensuring that we obtain near-optimal subsets efficiently.

### 3.3 A SINGLE-STOP SOLUTION FOR ALL FINE-TUNING STAGES

By combining the **utility-based kernel** with the above **submodular** objectives, we obtain DELIFT, a unified framework that selects data *holistically* across instruction tuning, task-specific fine-tuning, and continual learning:

- **Instruction Tuning**: Use Facility Location (FL) for diverse coverage of general instructions.
- **Task-Specific Fine-Tuning**: Use FLMI to align samples with a target benchmark or domain $\mathcal{D}_T$.
- **Continual Fine-Tuning**: Use FLCG to incorporate new, complementary data while avoiding redundancy with existing knowledge $\mathcal{D}_E$.

Despite these distinct objectives, each stage uses the same underlying *pairwise utility metric* to capture how one sample can enhance the model's predictions on another. This consistent, model-aware metric is key to DELIFT's strong performance across diverse tasks and model sizes.

## 4 EXPERIMENTAL RESULTS

We conduct extensive experiments to evaluate DELIFT in three fine-tuning scenarios: **(1) Instruction Tuning**, **(2) Task-Specific Fine-Tuning**, **(3) Continual Fine-Tuning**. This section outlines our experimental setup and baselines, followed by results across multiple models, datasets, and fine-tuning paradigms, along with interpretative commentary to utilize the extra space for deeper insights.

### 4.1 EXPERIMENTAL SETUP

**Models.** We evaluate DELIFT on both **base** LLMs (*Llama-3.2-3B*, *Mistral-7B-v0.1*, *opt-125m*) and **instruction-tuned** LLMs (*Qwen2-72B-Instruct*, *Phi-3-mini-128k-instruct*). This variety covers different parameter scales (millions vs. billions) along with diverse adaptation strategies (*ICL*, *QLoRA*, and, for smaller models, *full fine-tuning*). This diversity in model size and training setup provides a robust test-bed for assessing DELIFT across both resource-rich and more constrained environments.

**Datasets.** We group the datasets by the primary goal of fine-tuning, ensuring a clear mapping from the data to the corresponding submodular objective. In particular: **1. Instruction Tuning:**

*Mix-Instruct* (Jiang et al., 2023), *P3* (Sanh et al., 2021). Both aim to enhance general instruction-following behavior, featuring a variety of task prompts and user requests. **2. Task-Specific Fine-Tuning:** *HotpotQA* (Yang et al., 2018) aligned with *MMLU* (Hendrycks et al., 2021), *Mix-Instruct* aligned with *MT-Bench* (Zheng et al., 2023), and *Mix-Instruct* aligned with *GSM-8k* (Cobbe et al., 2021). These pairings allow us to extract only the most *relevant* samples from a large corpus to improve performance on a specific target benchmark. **3. Continual Fine-Tuning:** (a) *SQuAD* (Rajpurkar et al., 2016) paired with *HotpotQA* to inject more complex, multi-hop reasoning data after simpler QA, and (b) a proprietary IBM/Government domain query rewriting dataset.[1]

In all cases, we fixed an approximate budget of 30% for subset selection unless otherwise noted, striking a balance between data efficiency and coverage.

**Baselines.** In addition to a **Full Data** baseline, where the entire training set is used, we compare DELIFT with:

**1. Random**: Uniformly selects 30% of the training set to provide a simple, model-agnostic benchmark.

**2. SelectIT** (Liu et al., 2024): Generates self-reflection prompts within the LLM to rate data quality, filtering out lower-quality samples.

**3. LESS** (Xia et al., 2024): Employs gradient-based influence estimates, approximated via LoRA, to identify highly impactful data points for model parameter updates.

**4. DEFT-UCS**: Uses sentence embeddings to cluster the dataset for diversity; although it captures semantic variety, it lacks explicit model feedback to guide selection.

**4. DELIFT (SE)**: Operates the same submodular optimization as DELIFT but replaces our utility-based kernel with static sentence embedding similarities to highlight the benefit of DELIFT's dynamic, model-aware approach.

These baselines range from naive (Random) to sophisticated (gradient-based, reflection-based, or embedding-based), furnishing a thorough comparison against DELIFT.

**Metrics.** We measure model performance across different aspects:

**1. ROUGE** (Lin, 2004): Focuses on $n$-gram overlap for summarization tasks or generative text alignment.

**2. BGE** (Xiao et al., 2023): Evaluates semantic similarity through the dot product of normalized sentence embeddings.

**3. LAJ (LLM-as-Judge)** (Kim et al., 2023): Assigns a 1–5 rating reflecting correctness, clarity, and instruction adherence; we detail the scoring rubric and example prompts in Appendix C.

**4. Classification Accuracy**: Used primarily for multiple-choice tasks like *MMLU*.

By combining these metrics, we obtain both text-overlap measures (ROUGE), semantic evaluations (BGE), a holistic LLM-based score (LAJ), and a classification perspective (accuracy), offering a comprehensive view of model improvements under each fine-tuning scenario.

### 4.2 USE CASE 1: INSTRUCTION TUNING

**Setting.** Instruction tuning aims to broaden a model's capacity to follow instructions spanning diverse domains and styles. We adopt the **Facility Location (FL)** objective to maximize coverage of varied instruction types.

### 4.2.1 RESULTS ON INSTRUCTION-TUNED MODELS.

Tables 1 and 2 compare DELIFT with baselines on Mix-Instruct and P3, using two instruction-tuned models (**Qwen2-72B-Instruct** and **Phi-3-mini-128k-instruct**). Across both *ICL* and *QLoRA*,

---

[1]*Query rewriting* transforms follow-up queries (e.g., "How much is it?") into standalone forms (e.g., "How much is the subscription for IBM Cloud?"). This is essential for real-world systems where user queries often rely on prior context.

DELIFT surpasses other subset selection strategies and manages to prune up to 70% of data with minimal loss. Interestingly, DELIFT approaches—or occasionally exceeds—the Full Data baseline, indicating strong redundancy in instruction data.

| Model | Qwen2 | | | | | | Phi-3 | | | | | |
|---|---|---|---|---|---|---|---|---|---|---|---|---|
| Method | ICL | | | QLoRA | | | ICL | | | QLoRA | | |
| | ROUGE | BGE | LAJ | ROUGE | BGE | LAJ | ROUGE | BGE | LAJ | ROUGE | BGE | LAJ |
| Initial | 37.87 | 78.92 | 2.98 | 36.36 | 82.55 | 3.02 | 25.76 | 43.34 | 1.42 | 35.50 | 80.46 | 2.58 |
| Random | 39.00 | 80.66 | 3.12 | 44.45 | 85.46 | 3.12 | 33.05 | 72.73 | 2.92 | 44.70 | 83.75 | 2.95 |
| SelectIT | 43.08 | 84.50 | 3.18 | 45.14 | 85.88 | 3.21 | 36.11 | 76.31 | 3.18 | 49.68 | 85.84 | 3.20 |
| LESS | 42.08 | 83.24 | 3.26 | 45.16 | 84.95 | 3.28 | 47.10 | **85.94** | 3.23 | 48.68 | **85.86** | 3.24 |
| DELIFT (SE) | 47.43 | 84.40 | 3.28 | 48.22 | 86.50 | 3.28 | 46.62 | 85.28 | 3.24 | 45.64 | 83.70 | 3.27 |
| DELIFT | **48.46** | **85.77** | **3.35** | **52.79** | **88.04** | **3.37** | **49.83** | 85.27 | **3.32** | **50.31** | 84.40 | **3.33** |
| *Full Data* | *58.65* | *88.72* | *3.45* | *65.51* | *92.24* | *3.51* | *55.92* | *88.26* | *3.45* | *74.98* | *93.33* | *3.84* |

Table 1: Use Case 1: MixInstruct. **Bold** indicates best performance. DELIFT prunes 70% data yet stays close to Full Data.

| Model | Qwen2 | | | | | | Phi-3 | | | | | |
|---|---|---|---|---|---|---|---|---|---|---|---|---|
| Method | ICL | | | QLoRA | | | ICL | | | QLoRA | | |
| | ROUGE | BGE | LAJ | ROUGE | BGE | LAJ | ROUGE | BGE | LAJ | ROUGE | BGE | LAJ |
| Initial | 18.03 | 59.13 | 1.54 | 20.15 | 58.38 | 1.78 | 20.10 | 48.66 | 1.36 | 20.64 | 49.17 | 1.39 |
| Random | 20.05 | 59.39 | 1.79 | 20.29 | 59.39 | 1.83 | 20.83 | 49.92 | 2.24 | 24.51 | 53.41 | 2.36 |
| SelectIT | 31.38 | 71.08 | 2.86 | 32.96 | 74.76 | 2.90 | 35.37 | 66.67 | 2.52 | 38.98 | 69.84 | 2.54 |
| LESS | 34.59 | 83.23 | 3.07 | 35.03 | 83.37 | 3.50 | 39.69 | 72.12 | 3.17 | 40.32 | 70.89 | 3.24 |
| DELIFT (SE) | 34.69 | 83.31 | 3.43 | 35.46 | 83.43 | 3.53 | 37.07 | 71.49 | 3.52 | 38.13 | 79.68 | 3.74 |
| DELIFT | **35.48** | **83.69** | **3.58** | **35.60** | **83.64** | **3.54** | **40.66** | **84.00** | **3.68** | **41.91** | **84.53** | **3.76** |
| *Full Data* | *36.43* | *84.25* | *3.53* | *35.88* | *76.87* | *3.63* | *42.07* | *85.26* | *3.78* | *44.73* | *87.03* | *3.82* |

Table 2: Use Case 1: P3. DELIFT again discards most data while retaining strong performance.

### 4.2.2 EVALUATION ON A BASE MODEL.

Table 3 verifies these findings on a non-instruction-tuned **Llama-3.2-3B** (base). Even without prior instruction alignment, DELIFT outperforms gradient- and embedding-based baselines by a solid margin, corroborating that the utility metric is robust to model initialization.

| Method | ICL | | | QLoRA | | |
|---|---|---|---|---|---|---|
| | ROUGE | BGE | LAJ | ROUGE | BGE | LAJ |
| Initial | 25.88 | 61.97 | 1.90 | 28.51 | 73.14 | 2.48 |
| Random | 28.64 | 77.14 | 2.59 | 38.79 | 79.89 | 2.53 |
| SelectIT | 42.67 | 81.54 | 2.67 | 45.87 | 84.67 | 2.61 |
| DEFT-UCS | 41.55 | 80.12 | 2.63 | 41.03 | 81.86 | 2.59 |
| LESS | 44.99 | 82.48 | 2.69 | 50.54 | 84.14 | 2.78 |
| DELIFT (SE) | 51.19 | 83.54 | 2.72 | 55.32 | 85.92 | 2.79 |
| **DELIFT** | **54.58** | **88.29** | **2.83** | **58.57** | **90.98** | **2.98** |
| Full Data | 55.46 | 92.71 | 3.31 | 61.23 | 95.52 | 3.10 |

Table 3: Use Case 1: Llama-3.2-3B (base) on Mix-Instruct (30% subset).

**Takeaway (Use Case 1):** The FL objective and utility-based kernel consistently prune large volumes of instruction data without undermining final performance—a strong indicator that many instruction examples are either repetitive or uninformative.

### 4.3 USE CASE 2: TASK-SPECIFIC FINE-TUNING

**Setting.** We refine models for specialized domains (reasoning, QA). Using **Facility Location Mutual Information (FLMI)**, we select examples aligned with a target dataset $\mathcal{D}_T$.

| Method | Qwen2 (QLoRA) | Phi-3 (QLoRA) |
|---|---|---|
| Initial | 82.10 | 69.10 |
| Random | 79.31 | 65.16 |
| SelectIT | 79.13 | 65.24 |
| LESS | 80.35 | 66.72 |
| DELIFT (SE) | 80.10 | 66.36 |
| **DELIFT** | **81.70** | **68.70** |
| *Full Data* | *78.36* | *64.50* |

Table 4: Use Case 2: HotpotQA and MMLU (5-shot) for Qwen2 and Phi-3 (classification accuracy). DELIFT outperforms Full Data by 3.34% for Qwen2 and by 4.20% for Phi-3.

| Model | | Qwen2 | | | | | | Phi-3 | | | | |
|---|---|---|---|---|---|---|---|---|---|---|---|---|
| Method | ICL | | | QLoRA | | | ICL | | | QLoRA | | |
| | ROUGE | BGE | LAJ | ROUGE | BGE | LAJ | ROUGE | BGE | LAJ | ROUGE | BGE | LAJ |
| Initial | 44.32 | 74.86 | 2.48 | 47.65 | 77.92 | 2.72 | 39.57 | 69.43 | 2.31 | 42.89 | 72.76 | 2.53 |
| Random | 49.78 | 79.54 | 2.83 | 52.91 | 82.67 | 3.05 | 44.63 | 74.28 | 2.62 | 47.85 | 77.39 | 2.84 |
| SelectIT | 54.92 | 83.71 | 3.12 | 57.86 | 86.59 | 3.31 | 49.75 | 78.64 | 2.91 | 52.68 | 81.52 | 3.13 |
| LESS | 59.63 | 85.89 | 3.29 | 62.74 | 88.72 | 3.48 | 54.82 | 81.95 | 3.08 | 57.73 | 84.67 | 3.29 |
| DELIFT (SE) | 62.85 | 86.94 | 3.38 | 65.83 | 89.76 | 3.57 | 57.69 | 82.87 | 3.17 | 60.54 | 85.59 | 3.38 |
| DELIFT | **64.73** | **87.82** | **3.47** | **67.91** | **90.64** | **3.66** | **59.58** | **83.76** | **3.26** | **62.47** | **86.48** | **3.47** |
| *Full Data* | *65.89* | *88.65* | *3.55* | *69.72* | *91.53* | *3.74* | *60.76* | *84.59* | *3.34* | *64.31* | *87.42* | *3.55* |

Table 5: Use Case 2: Mix-Instruct and MT-Bench. **Bold** indicates the best performance. There is a 2.91% drop from Full Data to DELIFT after pruning 70% of the data, and a 1.14% drop from DELIFT to the next best baseline.

### 4.3.1 HotpotQA → MMLU & MixInstruct → MT-Bench.

**Empirical Findings.** Tables 5 and 4 illustrate DELIFT's consistent success in bridging the gap between diverse source datasets (Mix-Instruct or HotpotQA) and target benchmarks (MT-Bench, MMLU). In some cases (HotpotQA → MMLU), DELIFT surpasses the Full Data baseline by pruning unhelpful training points.

### 4.3.2 Further Experiments on Complex Reasoning Tasks.

To rigorously test DELIFT on more challenging datasets, we apply it to **Mistral-7B-v0.1 (base)** using **Mix-Instruct** aligned with **GSM-8k**, a corpus of math word problems that demand multi-step logical reasoning. Table 6 shows that DELIFT yields strong improvements under both ICL and QLoRA, surpassing all competing baselines. Even the simpler FL-only variant outperforms LESS and DEFT-UCS, highlighting the advantage of carefully selected examples when dealing with more complex reasoning tasks. Moreover, using FLMI for task alignment confers an additional boost, reinforcing the importance of matching the submodular objective to the dataset's complexity and problem-solving requirements.

| Method | ICL | | | QLoRA | | |
|---|---|---|---|---|---|---|
| | ROUGE | BGE | LAJ | ROUGE | BGE | LAJ |
| Initial | 17.06 | 55.13 | 1.35 | 27.95 | 65.92 | 1.36 |
| Random | 19.18 | 56.03 | 1.38 | 33.85 | 81.74 | 1.85 |
| SelectIT | 30.27 | 77.23 | 1.45 | 42.29 | 88.17 | 2.49 |
| DEFT-UCS | 30.06 | 76.99 | 1.59 | 41.45 | 87.84 | 2.08 |
| LESS | 31.69 | 77.87 | 2.26 | 43.86 | 88.22 | 2.60 |
| DELIFT (SE) | 31.84 | 78.62 | 2.44 | 43.04 | 90.53 | 2.54 |
| DELIFT (FL instead of FLMI) | 32.30 | 78.54 | 2.54 | 44.62 | 91.04 | 2.63 |
| **DELIFT** | **33.25** | **79.10** | **2.56** | **46.12** | **92.97** | **2.71** |
| Full Data | 35.33 | 81.59 | 2.79 | 49.10 | 94.57 | 2.85 |

Table 6: Use Case 2: **Mistral-7B-v0.1 (base)**: Mix-Instruct → GSM-8k with FLMI.

**Takeaway (Use Case 2):** By explicitly matching data to a target domain or benchmark, DELIFT discards irrelevant samples and—in some cases—yields better results than training on the entire dataset.

## 4.4 USE CASE 3: CONTINUAL FINE-TUNING

**Setting.** A model must assimilate new data without forgetting old knowledge. We use **Facility Location Conditional Gain (FLCG)** to prioritize novel samples while avoiding overlaps.

### 4.4.1 EXAMPLES: IBM → GOVERNMENT, SQUAD → HOTPOTQA.

Tables 7 and 8 confirm that DELIFT mitigates redundancy. Notably, on IBM → Government, DELIFT sees less than 1% drop vs. Full Data, versus a 3–4% gap for baselines like LESS and DEFT-UCS.

| Model | Qwen2 | | | | | | Phi-3 | | | | | |
|---|---|---|---|---|---|---|---|---|---|---|---|---|
| Method | ICL | | | QLoRA | | | ICL | | | QLoRA | | |
| | ROUGE | BGE | LAJ | ROUGE | BGE | LAJ | ROUGE | BGE | LAJ | ROUGE | BGE | LAJ |
| Initial | 44.11 | 70.49 | 2.43 | 48.49 | 80.85 | 2.62 | 40.66 | 58.68 | 1.52 | 43.96 | 69.56 | 2.29 |
| Random | 55.57 | 85.26 | 2.91 | 55.52 | 85.53 | 2.94 | 45.76 | 76.19 | 2.45 | 58.94 | 82.41 | 2.89 |
| SelectIT | 63.07 | 86.38 | 3.18 | 65.42 | 87.50 | 3.20 | 63.49 | 85.27 | 2.96 | 64.09 | 85.07 | 3.16 |
| LESS | 64.28 | 85.41 | 3.29 | 69.85 | 89.33 | 3.45 | 66.01 | 87.20 | 3.19 | 67.53 | 88.17 | 3.22 |
| DELIFT (SE) | 61.07 | 85.16 | 3.45 | 74.05 | **92.47** | 3.58 | 68.84 | 88.46 | 3.32 | 69.30 | 88.62 | 3.35 |
| DELIFT | **69.49** | **87.94** | **3.60** | **74.19** | 92.23 | **3.65** | **74.11** | **89.41** | **3.57** | **74.38** | **91.55** | **3.57** |
| *Full Data* | *66.08* | *87.84* | *3.65* | *76.83* | *92.63* | *3.74* | *71.23* | *91.10* | *3.52* | *77.12* | *91.10* | *3.64* |

Table 7: Use Case 3: IBM and Government. DELIFT effectively selects novel examples, outperforming baselines.

| Model | Qwen2 | | | | | | Phi-3 | | | | | |
|---|---|---|---|---|---|---|---|---|---|---|---|---|
| Method | ICL | | | QLoRA | | | ICL | | | QLoRA | | |
| | **ROUGE** | **BGE** | **LAJ** | **ROUGE** | **BGE** | **LAJ** | **ROUGE** | **BGE** | **LAJ** | **ROUGE** | **BGE** | **LAJ** |
| Initial | 51.51 | 66.97 | 1.77 | 54.18 | 78.27 | 2.50 | 40.42 | 58.23 | 1.26 | 40.94 | 58.12 | 1.29 |
| Random | 54.38 | 79.12 | 2.57 | 59.23 | 82.02 | 2.66 | 44.29 | 59.45 | 1.33 | 50.29 | 61.52 | 1.60 |
| SelectIT | 58.03 | 83.75 | 2.82 | 63.26 | 84.01 | 2.82 | 47.35 | 74.15 | 2.54 | 56.88 | 80.47 | 2.70 |
| LESS | 67.16 | 85.76 | 2.94 | 69.72 | 86.63 | 3.26 | 60.97 | 81.41 | 2.84 | 61.56 | 81.53 | 2.88 |
| DELIFT (SE) | 73.75 | 88.01 | 3.26 | 74.84 | 88.79 | 3.30 | 64.44 | 83.95 | 3.03 | 66.35 | 84.77 | 3.14 |
| DELIFT | **76.94** | **90.41** | **3.33** | **77.56** | **89.99** | **3.34** | **66.55** | **84.65** | **3.25** | **67.09** | **85.17** | **3.32** |
| *Full Data* | *77.78* | *90.31* | *3.35* | *78.72* | *90.77* | *3.48* | *68.47* | *85.93* | *3.33* | *70.48* | *86.06* | *3.44* |

Table 8: Use Case 3: SQuAD and HotpotQA. **Bold** indicates best performance. There is only a 1.94% drop from Full Data to DELIFT after pruning 70% of the data, and a 2.78% drop from DELIFT to the next best baseline.

**Takeaway (Use Case 3):** By prioritizing complementary rather than redundant data, DELIFT preserves previous capabilities and focuses on new insights.

## 4.5 FURTHER COMPARISONS AND ABLATIONS

### 4.5.1 FULL FINE-TUNING VS. QLORA.

We also validated DELIFT under *full fine-tuning* on a smaller **opt-125m** (Table 9). Results show consistent gains for DELIFT, indicating that the utility-based selection approach is independent of the particular fine-tuning methodology.

### 4.5.2 COMPARING SUBMODULAR OBJECTIVES.

Although FL alone can beat naive baselines in specialized or incremental settings (see Table 6), the specialized objectives (FLMI for domain tasks, FLCG for continual updates) yield stronger alignment. This underscores the importance of matching the submodular objective to the fine-tuning stage.

| Method | QLoRA | | | Full Fine-Tuning | | |
|---|---|---|---|---|---|---|
| | ROUGE | BGE | LAJ | ROUGE | BGE | LAJ |
| Initial | 9.04 | 40.50 | 1.19 | 9.57 | 40.86 | 1.14 |
| Random | 12.55 | 46.99 | 1.25 | 13.07 | 47.91 | 1.30 |
| SelectIT | 14.78 | 49.80 | 1.26 | 15.35 | 50.42 | 1.28 |
| DEFT-UCS | 15.12 | 50.29 | 1.39 | 15.16 | 50.29 | 1.39 |
| LESS | 15.52 | 50.70 | 1.38 | 16.02 | 51.30 | 1.40 |
| DELIFT (SE) | 18.81 | 55.02 | 1.35 | 17.06 | 55.93 | 1.38 |
| **DELIFT** | **19.72** | **57.98** | **1.43** | **19.87** | **58.11** | **1.45** |
| *Full Data* | *20.39* | *60.39* | *1.95* | *21.64* | *61.70* | *2.05* |

Table 9: **opt-125m** on Mix-Instruct, comparing QLoRA vs. full fine-tuning.

### 4.5.3 ABLATION ON SUBSET SIZE.

Beyond consistently using 30% of the data in our main experiments, we investigated how varying the subset size influences performance. We tested budgets ranging from as little as 5% up to 50% of the original training set (in increments of 10%). As shown in Figure 2, performance under all methods generally improves with larger subsets but exhibits diminishing returns beyond 30–40%. Notably, DELIFT consistently yields higher LAJ scores than other baselines at every subset size, demonstrating its robustness even under very aggressive pruning (e.g., 5%). These findings reinforce that substantial reductions in training data are possible without significant performance degradation, and that DELIFT identifies highly informative samples more effectively than competing methods.

### 4.6 KEY OBSERVATIONS

Our experiments reveal four consistent patterns across datasets, model scales, and adaptation methods:

1. **Utility-based kernel outperforms static or gradient-based methods**, indicating that measuring how one sample improves predictions on another is a stronger signal than mere embeddings or approximate gradients.
2. **Stage-specific objectives (FL, FLMI, FLCG)** are crucial for addressing the unique needs of instruction tuning (diverse coverage), task-specific alignment, and continual complementarity.
3. **Significant pruning (up to 70%)** is routinely possible, often with minimal or zero performance deterioration, suggesting vast redundancy in large corpora.
4. **Method-agnostic gains**, as DELIFT consistently improves results under ICL, QLoRA, and even full fine-tuning on smaller models.

In the next section, we summarize these findings, discuss limitations such as the $O(n^2)$ cost of utility computation, and propose directions for future work, including adaptive selection during training, extensions to multimodal data, and fairness considerations.

## 5 CONCLUSION, LIMITATIONS, AND FUTURE WORK

In this paper, we introduced DELIFT, a novel approach to data-efficient fine-tuning of large language models by employing a versatile pairwise utility metric combined with submodular optimization techniques for optimal data selection. Empirical evaluations showed that DELIFT can reduce data and computational requirements by up to 70% while achieving performance comparable to the full dataset, and outperforming existing data selection methods by up to 26% in effectiveness. These results suggest that DELIFT offers a promising method for improving the accessibility of LLM adaptation, especially for resource-constrained scenarios. However, our approach has limitations, including potential sensitivity to the quality and diversity of initial data and the risk of bias amplification inherent in the selected data. Future work will explore integrating DELIFT with data augmentation techniques to improve robustness, incorporating fairness constraints to mitigate biases, and extending the approach to emerging model architectures and multimodal learning. Our ongoing efforts are directed toward ensuring that DELIFT contributes to responsible and equitable AI development while maximizing efficiency.

## 6 ACKNOWLEDGEMENT

A part of this work used the Delta system at the National Center for Supercomputing Applications through allocation CIS240550 from the Advanced Cyberinfrastructure Coordination Ecosystem: Services & Support (ACCESS) program, which is supported by National Science Foundation grants #2138259, #2138286, #2138307, #2137603, and #2138296.

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

# Appendix

## A   THEORETICAL FOUNDATIONS AND CONNECTIONS BETWEEN THE UTILITY METRIC AND INFORMATION THEORY

**Theorem 2** *Let $y_i = (y_{i1}, y_{i2}, \ldots, y_{iT})$ be a sequence of tokens with ground truth distribution $GT_i$, where $GT_i$ assigns probability 1 to the sequence $y_i$ and 0 to all other sequences. Let $p(y_i \mid x_i)$ be the predicted probability of $y_i$ given input $x_i$, and $p(y_i \mid x_i, x_j, y_j)$ be the predicted probability of $y_i$ given $x_i$ and an in-context example $(x_j, y_j)$. Define the utility metric $UF_{ij}$ using a general distance metric $d(\cdot, \cdot)$ between probability distributions:*

$$UF_{ij} = d(GT_i, \, p(y_i \mid x_i)) - d(GT_i, \, p(y_i \mid x_i, x_j, y_j)).$$

***Claim:*** *When the distance metric $d(\cdot, \cdot)$ is the Kullback-Leibler divergence $D_{KL}$, the utility metric $UF_{ij}$ is equal to the pointwise mutual information (PMI) between the sequence $y_i$ and the in-context example $(x_j, y_j)$ conditioned on $x_i$:*

$$UF_{ij} = \mathrm{PMI}(y_i; \, x_j, y_j \mid x_i) = \log \frac{p(y_i \mid x_i, x_j, y_j)}{p(y_i \mid x_i)}.$$

*Furthermore, $UF_{ij}$ can be expressed as the sum of conditional PMI over the tokens in $y_i$:*

$$UF_{ij} = \sum_{t=1}^{T} \mathrm{PMI}\left(y_{it}; \, x_j, y_j \mid x_i, y_{i,<t}\right),$$

*where $y_{i,<t} = (y_{i1}, y_{i2}, \ldots, y_{i(t-1)})$ denotes the sequence of previous tokens up to position $t - 1$.*

***Proof:***

*1. Computing KL-Divergence Between Ground Truth and Predicted Distributions:*

*Since $GT_i$ assigns probability 1 to the specific sequence $y_i$, the KL-divergence simplifies as follows:*

$$d(GT_i, \, p(y_i \mid \cdot)) = D_{KL}(GT_i \parallel p(y_i \mid \cdot)) = -\log p(y_i \mid \cdot),$$

*because the KL-divergence between a one-hot distribution and any other distribution reduces to the negative log-probability of the assigned event.*

*2. Computing the Utility Metric $UF_{ij}$:*

*The utility metric becomes:*

$$\begin{aligned}
UF_{ij} &= -\log p(y_i \mid x_i) + \log p(y_i \mid x_i, x_j, y_j) \\
&= \log \frac{p(y_i \mid x_i, x_j, y_j)}{p(y_i \mid x_i)}.
\end{aligned}$$

*3. Expressing $UF_{ij}$ as Pointwise Mutual Information:*

*The conditional pointwise mutual information between $y_i$ and $(x_j, y_j)$ given $x_i$ is defined as:*

$$\mathrm{PMI}(y_i; \, x_j, y_j \mid x_i) = \log \frac{p(y_i, x_j, y_j \mid x_i)}{p(y_i \mid x_i) \, p(x_j, y_j \mid x_i)}.$$

*Using the chain rule:*

$$p(y_i, x_j, y_j \mid x_i) = p(y_i \mid x_i, x_j, y_j) \, p(x_j, y_j \mid x_i).$$

*Substituting back:*

$$\text{PMI}(y_i; \, x_j, y_j \mid x_i) = \log \frac{p(y_i \mid x_i, x_j, y_j) \, p(x_j, y_j \mid x_i)}{p(y_i \mid x_i) \, p(x_j, y_j \mid x_i)}$$

$$= \log \frac{p(y_i \mid x_i, x_j, y_j)}{p(y_i \mid x_i)}.$$

*Therefore:*

$$UF_{ij} = \text{PMI}(y_i; \, x_j, y_j \mid x_i).$$

*4. Expanding $UF_{ij}$ as Sum of Conditional PMI Terms:*

*We expand $p(y_i \mid \cdot)$ using the chain rule:*

$$p(y_i \mid \cdot) = \prod_{t=1}^{T} p(y_{it} \mid \cdot, y_{i,<t}),$$

*where $y_{i,<t}$ is the sequence of previous tokens up to time $t - 1$.*

*Substituting back into $UF_{ij}$:*

$$UF_{ij} = \log \frac{\prod_{t=1}^{T} p(y_{it} \mid x_i, x_j, y_j, y_{i,<t})}{\prod_{t=1}^{T} p(y_{it} \mid x_i, y_{i,<t})}$$

$$= \sum_{t=1}^{T} \left[ \log p(y_{it} \mid x_i, x_j, y_j, y_{i,<t}) - \log p(y_{it} \mid x_i, y_{i,<t}) \right]$$

$$= \sum_{t=1}^{T} \text{PMI}\left(y_{it}; \, x_j, y_j \mid x_i, y_{i,<t}\right).$$

*This shows that $UF_{ij}$ is the sum of the conditional PMI of each token $y_{it}$ with $(x_j, y_j)$ given $x_i$ and the previous tokens.*

### Conclusion:

*When $d(\cdot, \cdot) = D_{KL}$, the utility metric $UF_{ij}$ precisely equals the conditional PMI between $y_i$ and $(x_j, y_j)$ given $x_i$.*

### A.1 Why Euclidean Distance is Preferred Over KL-Divergence for Subset Selection

The effectiveness of subset selection algorithms, including facility location functions, depends critically on the properties of the chosen distance metric $d(\cdot, \cdot)$. Euclidean distance offers several key advantages over KL-divergence for this purpose:

1. **Mathematical Properties**
   - Euclidean distance is non-negative, finite, and symmetric (d(a,b) = d(b,a))
   - KL-divergence can be infinite or undefined with zero probabilities and lacks symmetry $D_{KL}(P \parallel Q) \neq D_{KL}(Q \parallel P)$

2. **Computational Advantages**
   - Euclidean distance uses simple arithmetic operations (subtraction, squares, square roots)
   - KL-divergence requires more complex logarithmic calculations and division operations

3. **Robustness in Practice**
   - Euclidean distance handles zero probabilities gracefully
   - KL-divergence becomes undefined with zero probabilities, which occur frequently in real data

**Impact on Subset Selection:** The facility location function requires positive, finite similarity measures to model coverage effects accurately. Euclidean distance satisfies these requirements, while KL-divergence's potential negative or infinite values can disrupt optimization.

**Conclusion:** While KL-divergence offers theoretical connections to mutual information, Euclidean distance provides:

- Guaranteed positive and finite utility metrics
- Superior computational efficiency
- Better numerical stability

These practical advantages make Euclidean distance the preferred choice for computing the utility metric $UF_{ij}$ in subset selection algorithms.

# B    SUBSET SIZE COMPARISON

To assess how subset size influences the performance of DELIFT, we performed an ablation study by varying the subset size from 5% to 100% (specifically 5%, 15%, 30%, 50%, 100%) of the entire dataset across three distinct use cases. Figure 2 illustrates the performance metric LAJ as a function of subset size for each fine-tuning scenario.

## B.1    GENERAL OBSERVATIONS

- **Performance Increases with Subset Size:** Across all methods, LAJ scores generally improve as the subset size increases. Utilizing the full dataset consistently yields the highest performance, underscoring the benefits of a larger training set.
- **Diminishing Returns Beyond 50%:** For methods such as DELIFT, SelectIT, and LESS, performance gains plateau or slow down beyond a 50% subset size. This suggests that additional data beyond this point offers minimal benefits and may introduce redundancy.

## B.2    DETAILED ANALYSIS OF METHODS

### B.2.1    INITIAL VS. RANDOM SELECTION

- **Initial Baseline:** Consistently records the lowest scores across all subset sizes, indicating that models without data-informed selection struggle to generate quality responses.
- **Random Selection:** Slightly outperforms the Initial baseline but maintains a relatively flat performance curve. This lack of significant improvement highlights that uninformed data selection does not substantially enhance model quality.

### B.2.2    SELECTIT AND LESS METHODS

- **LESS:** Demonstrates a strong upward trend, particularly when subset sizes increase from 15% to 50%. This indicates that LESS effectively selects informative subsets, especially in the mid-range subset sizes, but is sub-optimal with smaller subset sizes.
- **SelectIT:** Initially lags behind DELIFT and LESS but shows steady improvement with larger subset sizes. For subset sizes above 50%, SelectIT approaches the performance of DELIFT, suggesting its heuristic-driven selection becomes more effective with more data.

### B.2.3    DELIFT VARIANTS

- **DELIFT vs. DELIFT (SE):** DELIFT consistently outperforms DELIFT (SE), which uses sentence embeddings, highlighting the superiority of DELIFT's utility-based kernel in capturing data informativeness.

- **DELIFT vs. Other Methods:** DELIFT outperforms all other subset selection methods across all subset sizes, particularly below 50%. This effectiveness is attributed to DELIFT's strategy of identifying the most informative samples early on, making it ideal for scenarios with limited computational resources.
- **DELIFT vs. Full Data:** At smaller subset sizes (e.g., 15%, 30%), DELIFT achieves LAJ scores close to the Full Data baseline. In ICL fine-tuning scenarios, a 30% subset size with DELIFT nearly matches Full Data performance, demonstrating its efficiency in data reduction without significant loss in performance.

### B.3 Impact on Different Fine-Tuning Scenarios

- **ICL vs. QLoRA:** QLoRA fine-tuning generally yields higher scores than ICL across all methods, suggesting that QLoRA benefits more from effective data selection strategies. DELIFT, in particular, shows more pronounced improvements in QLoRA settings, indicating its subsets are well-suited for efficient parameter tuning.
- **Use Case Comparisons:** In Use Case 3 (IBM and Government datasets), DELIFT achieves the highest gains relative to the Initial baseline across both ICL and QLoRA scenarios. This effectiveness is likely due to the nature of query rewriting tasks, where DELIFT's informed data selection effectively eliminates redundant or irrelevant examples, resulting in a higher-quality training set.

## C  Prometheus Rubric

The Prometheus model served as an LLM-as-a-Judge to evaluate response quality from different data selection methods. Table 10 contains the general rubric used for the Prometheus model scoring on all use cases and settings (except for the experiments on the query-rewriting task using the IBM-proprietary data).

### C.1 Usage Notes

- Each response is evaluated independently based on the criteria above.
- The cumulative score reflects the overall quality and effectiveness of the response.
- Final LAJ scores are obtained by averaging the scores across all criteria.

## D  LLM-as-Judges Scores

In Tables 11 and 12, we show the distribution of Prometheus scores on one particular setting: Use Case 1, MixInstruct training and MixInstruct validation sets on the Qwen2-72B-Instruct model. These figures make clear that the average LGA scores computed in Tables 1-8 are true averages of a distribution of scores, not averages of a combination of just 1's and 5's.

### D.1 Interpretation of Score Distributions

#### D.1.1 Overall Trends

- **Score Variability:** There is significant variability in score distributions across different methods. The Initial and Random baselines show a concentration of scores between 2.5 and 3.5, indicating average to subpar performance.
- **Enhanced Performance with Advanced Methods:** Methods like SelectIT, LESS, DELIFT (SE), and DELIFT exhibit score distributions skewed towards higher values (3.5 to 4.0), with DELIFT showing the highest concentration above 3.5. This highlights their effectiveness in selecting informative and useful data for fine-tuning.

#### D.1.2 Method-Specific Observations

- **Initial and Random Methods:** Both methods have lower mean scores (around 3.0 to 3.2) with wide spreads, suggesting inconsistent and generally lower-quality responses.

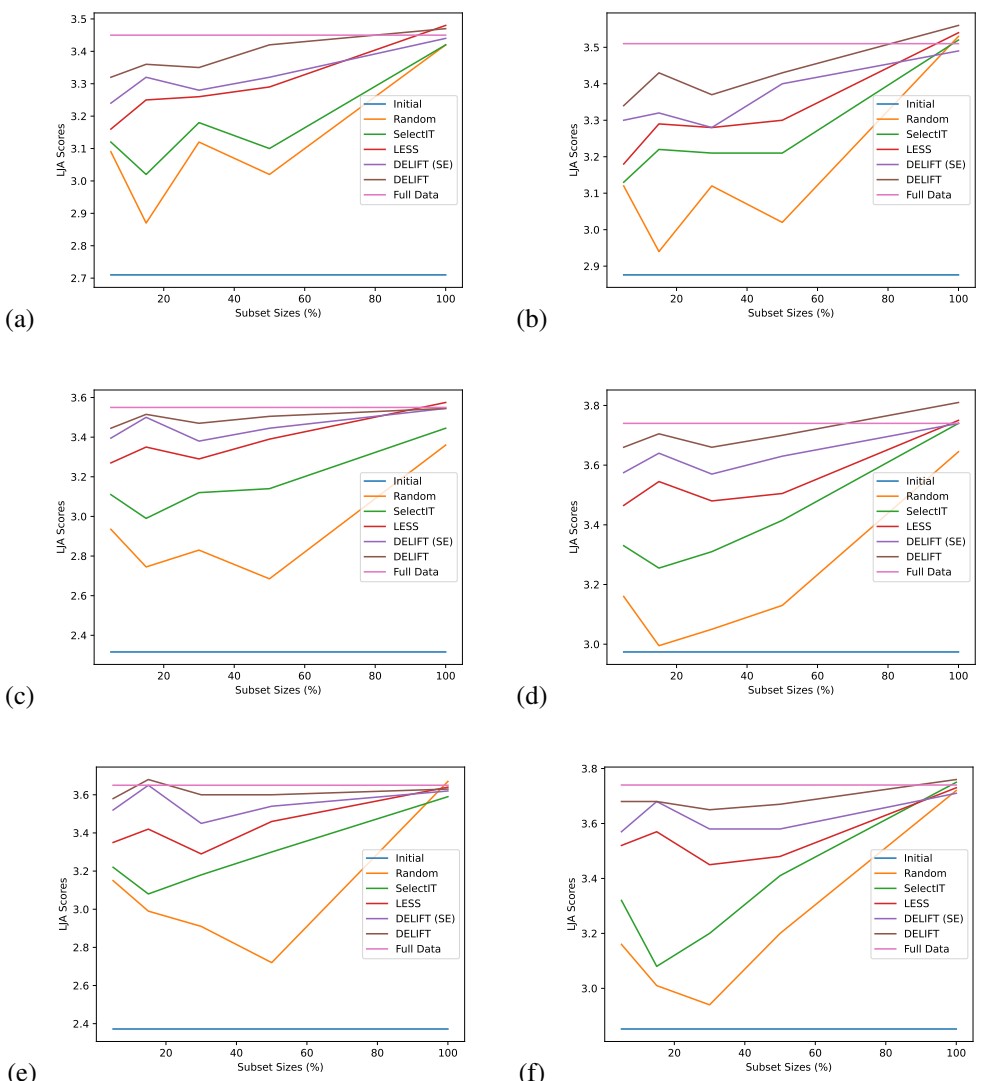

Figure 2: Graphs of LLM-A-J scores (y-axis) of Qwen2-72B-Instruct with varying subset sizes (x-axis) of Use Case 1 on MixInstruct for **(a)** ICL and **(b)** QLoRA, Use Case 2 on MixInstruct and MT-Bench for **(c)** ICL and **(d)** QLoRA, and Use Case 3 on IBM and Government for **(e)** ICL and **(f)** QLoRA.

Evaluate the model's ability to follow instructions and deliver a high-quality response across the following dimensions:
1. **Instruction Following**: How accurately and fully does the model adhere to the given instruction?
2. **Accuracy**: Is the information correct, reliable, and factually sound?
3. **Relevance**: Does the response directly address the question or task without unnecessary information?
4. **Completeness**: Does the response cover all essential aspects of the instruction or question
5. **Depth**: How thoroughly does the response explore the topic? Does it demonstrate insightful analysis where appropriate?
6. **Clarity**: Is the response well-organized, easy to follow, and free from ambiguity or confusion?
7. **Creativity**: Does the response offer original or innovative approaches where applicable?
8. **Helpfulness**: Does the response effectively meet the user's needs and provide value in solving the problem or addressing the query?

**Score of 1**: The response fails to meet expectations across most or all criteria. It does not follow the instruction, contains significant errors or misinformation, lacks relevance, is incomplete or shallow, unclear, unoriginal, and unhelpful.
**Score of 2**: "The response shows major deficiencies across several criteria. It partially follows the instruction but includes significant inaccuracies, is often irrelevant, incomplete, or lacks depth, clarity, creativity, and helpfulness.
**Score of 3**: "The response is average, meeting some but not all criteria. It follows the instruction but may fall short in terms of accuracy, depth, relevance, or helpfulness. Improvements in clarity and insightfulness may be needed.
**Score of 4**: The response is strong, performing well across most criteria. It follows the instruction closely, is mostly accurate and relevant, provides good depth, and is well-structured. Minor improvements could enhance clarity, creativity, or helpfulness.
**Score of 5**: "The response excels in all or nearly all criteria. It fully follows the instruction, is highly accurate, directly relevant, complete, and demonstrates depth and insight. The response is well-organized, creative where appropriate, and very helpful in addressing the user's needs.

Table 10: General Prometheus Rubric

- **SelectIT and LESS Methods:**

  - **SelectIT:** Shows improved mean scores, especially in QLoRA settings, indicating its effectiveness in resource-constrained training scenarios.

  - **LESS:** Demonstrates significant performance improvements, with mean scores around 3.26 to 3.28, reflecting effective gradient-based data selection.

- **DELIFT Variants:**

  - **DELIFT (SE):** Skews towards higher scores but not as prominently as DELIFT.

  - **DELIFT:** Achieves the highest average scores (3.35 for ICL and 3.37 for QLoRA), outperforming all other methods and indicating its superior utility-based kernel and submodular optimization.

### D.1.3 COMPARISON WITH FULL DATA

- **DELIFT vs. Full Data:** DELIFT nearly matches Full Data performance with only a slight reduction in mean scores (3.35 to 3.37 vs. 3.45 to 3.51). This demonstrates DELIFT's capability to retain most of the model's performance while using significantly less data.

- **Efficiency of Data Pruning:** Full Data shows a modest increase in mean scores compared to DELIFT, but at the cost of substantially higher computational resources. DELIFT offers a more efficient alternative without major sacrifices in performance.

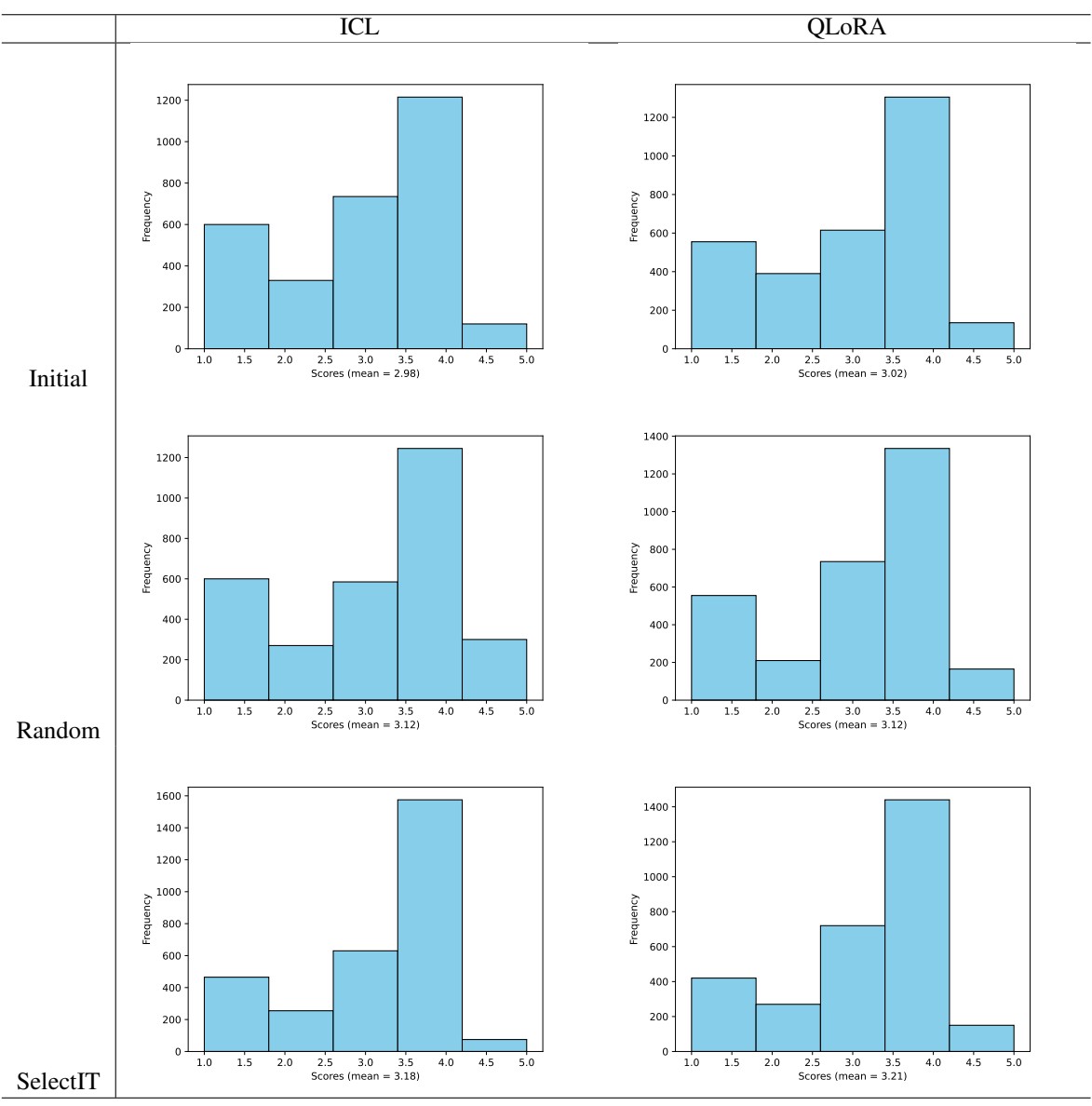

Table 11: LLM-as-Judges score distributions for Use Case 1 with MixInstruct training and validation set on the Qwen2-72B-Instruct model on the Initial, Random, and SelectIT baselines. The corresponding table is Table 1.

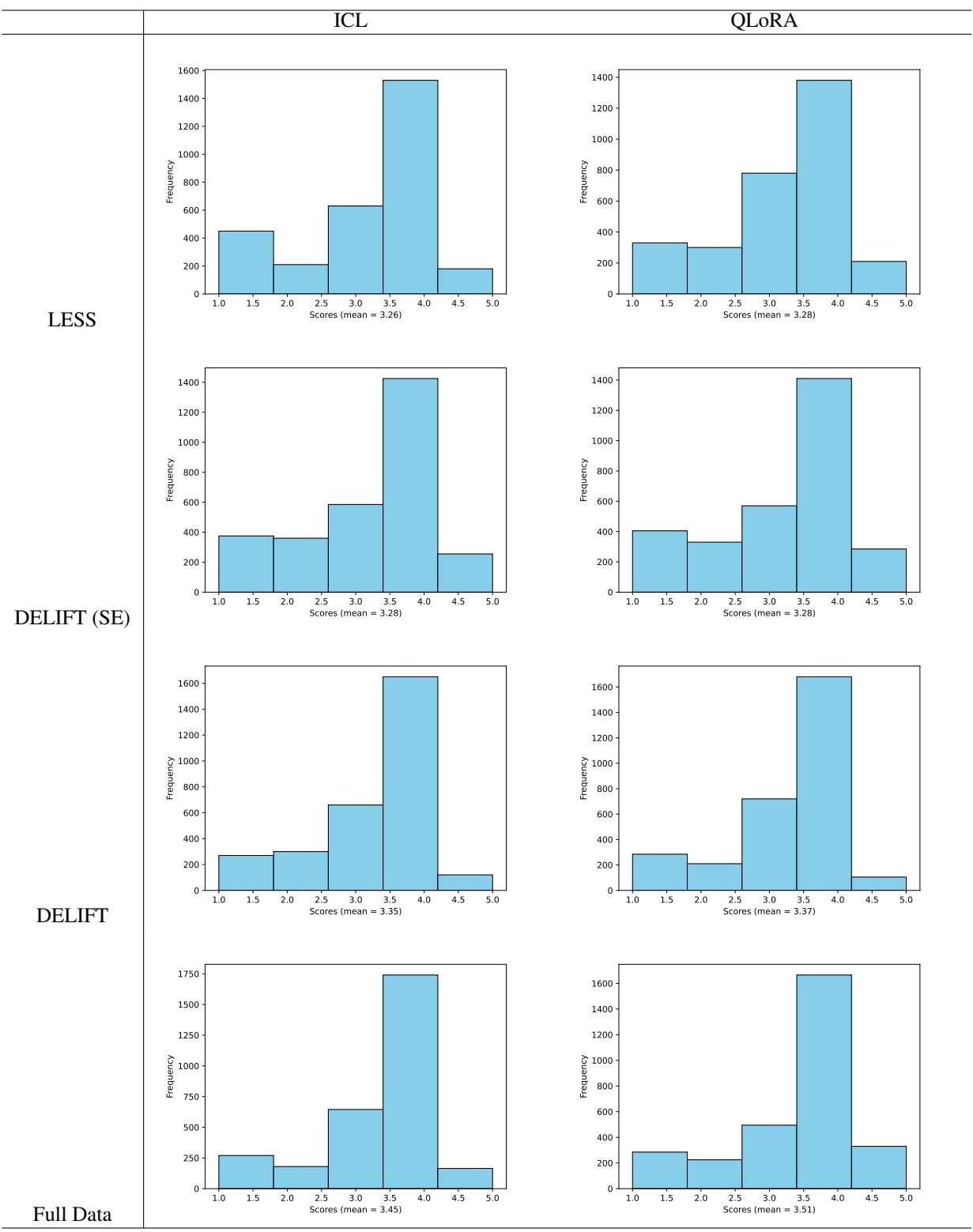

Table 12: LLM-as-Judges score distributions for Use Case 1 with MixInstruct training and validation set on the Qwen2-72B-Instruct model on the LESS, DELIFT with Sentence Embedding, DELIFT, and Full Data methods. The corresponding table is Table 1.

## E  LIMITATIONS

- **Dependence on Initial Data Quality:** DELIFT's effectiveness relies on the diversity and quality of the initial dataset. Biases or lack of diversity in the dataset can propagate to the selected subsets.

- **Scalability Constraints:** While DELIFT is computationally efficient, extremely large datasets may still present challenges in terms of computation and memory.

- **Domain-Specific Performance:** DELIFT's performance may vary across different domains, particularly those requiring specialized knowledge or handling multimodal data.

- **Bias Amplification Risks:** The subset selection process may unintentionally amplify existing biases within the data, necessitating careful mitigation strategies.

## F  FUTURE WORK

- **Integration with Data Augmentation:** Combining DELIFT with data augmentation techniques could further enhance the robustness and diversity of selected subsets.

- **Fairness and Bias Mitigation:** Incorporating fairness constraints and bias mitigation strategies into the subset selection process to ensure equitable model performance across different groups.

- **Extension to Multimodal Learning:** Adapting DELIFT for multimodal data (e.g., text, images, audio) to expand its applicability beyond natural language processing.

- **Theoretical Analysis:** Developing a deeper theoretical understanding of the utility metric and its properties to further validate and refine the approach.

- **Enhancing Scalability:** Exploring methods to scale DELIFT effectively for larger datasets and more complex models without compromising efficiency.

Our ongoing efforts aim to ensure that DELIFT contributes to responsible and equitable AI development while maximizing efficiency.

## G  CODE AND DATA AVAILABILITY

To facilitate reproducibility and further research, we will make the DELIFT implementation and the datasets used in our experiments publicly available. Interested researchers can access these resources through the following repository: https://anonymous.4open.science/r/optimizing-data-selection-0CD0.

## H  HYPERPARAMETER SETTINGS

Consistent hyperparameter settings were maintained across all experiments to ensure reproducibility:

- **Submodular Function:** Utilized Facility Location (FL), Facility Location Mutual Information (FLMI), or Facility Location Conditional Gain (FLCG) based on the use case.

- **Utility Metric Scaling Factor:** Set $\eta = 1$ for FLMI and $\nu = 1$ for FLCG.

- **Budget (% of Data):** Fixed at 30% for all subset selection experiments.

- **Optimization Algorithm:** Employed greedy maximization with a stopping criterion based on the budget.

- **Distance Metric:** Used length-normalized L2 norm.

- **Teacher Forcing Technique:** Applied during utility metric computation to ensure reliable prediction accuracy measurement.

$$\text{corr}_{AB} = \sum_{i \in B} p(y_i|x_i)_{M_A} - p(y_i|x_i)_{BM} \qquad (2)$$

