# OpenReview forum: "DELIFT: Data Efficient Language model Instruction Fine-Tuning"
_ICLR.cc/2025/Conference — ICLR 2025 Poster_

### Official Review · Reviewer_QaUn · 2024-10-30

**Soundness:** 3
**Presentation:** 2
**Contribution:** 3
**Rating:** 6
**Confidence:** 3

**Summary:**

In this paper, the authors propose a data selection method named DELIFT. DELIFT first employs a pairwise utility metric to quantify how beneficial a data sample is for improving the model's responses to other samples. Basing on the pairwise utility metric, DELIFT leverages 3 specially designed submodular functions to conduct diverse subsets that are useful across the instruction tuning, task-specific fine-tuning and continual fine-tuning. Experiments across various tasks and model scales demonstrate that DELIFT can reduce the fine-tuning data size by up to 70% without compromising performance, offering significant computational savings and outperforming existing methods in both efficiency and efficacy.

**Strengths:**

1. **Effective performance of DELIFT** In this paper, the authors design a utility-based kernel to quantify the informativeness of data. Based on this, the authors further design submodular functions for instruction data selection, targeting Instruction Tuning, Task-specific Fine-Tuning, and Continual Fine-Tuning. This method has been proven effective in all three stages and outperforms other model-dependent baselines.

**Weaknesses:**

1. **Related Work not Discussed** In Entropy Law [1], the authors also addressed the instruction data selection problem from the perspective of informativeness (entropy) and quantifies the relationship between data compression ratio and model performance; However, [1] is not discussed in this paper.

[1] Entropy Law: The Story Behind Data Compression and LLM Performance

**Questions:**

See weakness

---

> ### Author Response · Authors · 2024-11-18
>
> Thank you for your detailed review and for highlighting the strengths of DELIFT. We also appreciate the suggestion to discuss the "Entropy Law" paper. This paper provides an interesting perspective by leveraging compression measures to identify optimal subsets, which is somewhat analogous to using semantic embeddings with the Facility Location function in our method.
>
> However, our approach differs fundamentally in several ways:
>
> 1. **Utility vs. Compression-Based Measures**: While the entropy-based method identifies representative subsets using bit compression metrics, such measures focus primarily on the compactness and redundancy of the data. Similarly, semantic embedding-based approaches capture surface-level diversity but often fail to account for the utility of a sample in enhancing model performance. In contrast, DELIFT directly employs a utility-based kernel to evaluate the contribution of a sample to downstream performance, resulting in empirically superior outcomes (e.g., as demonstrated in our results comparing DELIFT to DELIFT (SE)).
>
> 2. **Theoretical and Empirical Strength of Facility Location**: The Facility Location function used in DELIFT has well-established theoretical guarantees, including provable approximation bounds for submodular maximization, ensuring the selection of diverse and representative subsets. Empirically, our results further validate that the subsets selected using DELIFT outperform those chosen by methods relying on compression or semantic embeddings in terms of both efficiency and efficacy.
>
> 3. **Versatility Across Fine-Tuning Stages**: Another key distinction is the scope of application. While the entropy-based approach targets the selection of representative subsets from a dataset, it does not offer the flexibility to address specific needs such as selecting new, complementary samples for continual fine-tuning. DELIFT, on the other hand, is specifically designed to optimize subset selection across diverse fine-tuning stages, including instruction tuning, task-specific fine-tuning, and continual fine-tuning, ensuring both theoretical robustness and practical applicability.
>
> We have incorporates these clarifications into the revised related work section to better position DELIFT in the context of related approaches.
>
> ---
>
> We thank the reviewer for their time. Please let us know of any more questions and concerns you might have - we are happy to answer them. If we addressed your concerns sufficiently, we would appreciate to see this reflected in an increase of rating scores. Thank you again!

---

> ### Author Response · Authors · 2024-11-22
>
> We thank the reviewer for their thoughtful feedback and for taking the time to review our work. If there are any remaining concerns or additional points you'd like us to address, we would be happy to do so. If our revisions have sufficiently addressed your feedback, we would appreciate your consideration of this in your evaluation. Thank you again for your time and effort!

---

### Official Review · Reviewer_hvNC · 2024-11-03

**Soundness:** 3
**Presentation:** 3
**Contribution:** 3
**Rating:** 6
**Confidence:** 3

**Summary:**

This paper addresses the challenge of selecting optimal data for fine-tuning large language models (LLMs). The authors propose a pairwise utility metric to assess the value of each sample based on the model's current performance and the relevance of other samples in the dataset. They introduce three variations of a facility location (FL) function to guide data selection at different fine-tuning stages, including (i) a vanilla FL function to prioritize diversity in general instruction fine-tuning, (ii) a mutual information-based FL function to select data that is diverse and relevant to a target task for task-specific fine-tuning, and (iii) a conditional gain-based FL function to select data that complements previously used data for continual fine-tuning. Experiments on two instructional models, Phi-3-mini and Qwen2-72B, demonstrate that the proposed methods outperform existing data selection techniques across all three fine-tuning stages.

**Strengths:**

- Data selection for LLM fine-tuning is a timely research problem, particularly in light of the growing model sizes, data availability, and resource constraints.
- The paper discusses the varying criteria for data selection across fine-tuning stages. The proposed method can be customized for different fine-tuning objectives.

**Weaknesses:**

- The utility metric in Equation 1 is calculated based on in-context learning (ICL) gains, which may not directly translate to improvements in fine-tuning performance. Is there supporting literature to justify the use of ICL-derived utility in this context?
- It remains unclear how the utility gain measured through ICL reflects sample diversity in the semantic space. Since utility values are computed only once at the beginning of the data selection process, they may not capture changes in the model's performance after fine-tuning on the current selected subset.
- The experimental design would benefit from comparisons with additional diversity-based baselines (e.g., clustering). Analysis of the selected examples across different methods would provide insights into the strengths and weaknesses of each approach.
- A more thorough ablation analysis on the FL functions is needed. Do the additional terms for each fine-tuning stage significantly contribute to the results? Would the vanilla FL perform adequately across all stages? Evaluating this would help clarify the need for distinct data selection strategies at each fine-tuning stage.

**Questions:**

- In equation 3-5, what is $s_ij$? Is it the utility-based kernel $UF_ij$? Is it recomputed after each selection round?
- For Equation 1, how is the ground truth distribution determined? The description of "a vector of ones for each token to signify perfect predictions" needs further clarification.
- The latex template of the submitted manuscript appears modified, as it lacks line numbers in the left margin.
- Why are instruction-finetuned models used for the experiments rather than pre-trained models, especially given that the first use case is instruction fine-tuning?

---

> ### Author Response · Authors · 2024-11-18
>
> Thank you for your response and the appreciation of our solution towards a timely research problem.
>
> **Q1: "ICL gains and fine-tuning improvements"**
>
> **A1.** Our utility metric \(UF_{ij} \) is grounded in a well-established framework based on the distance between probability distributions. Specifically, for a ground truth distribution \( GT_i \) and the model's predicted probabilities \( p(y_i | x_i) \), we define the utility metric as:
>
> $
> UF_{ij} = d(GT_i,\, p(y_i \mid x_i)) - d(GT_i,\, p(y_i \mid x_i, x_j, y_j)),
> $
>
> where $d(\cdot, \cdot)$ represents the distance metric between the distributions;  $GT_i$ assigns probability 1 to the sequence $y_i$ and 0 to all other sequences; $p(y_i \mid x_i)$ is the model's predicted probability for $y_i$ given only the input $x_i$; and $p(y_i \mid x_i, x_j, y_j)$ is the predicted probability for $y_i$ when the model is provided with $(x_j, y_j)$ as an in-context example along with $x_i$.
>
> **Theoretically**, when using Kullback-Leibler (KL) divergence as the distance metric $d$, we prove that $UF_{ij}$ is equal to the conditional pointwise mutual information (PMI) between $y_i$ and $(x_j, y_j)$ given $x_i$. Specifically, we have:
>
> $$
> UF_{ij} = \operatorname{PMI}(y_i;\, x_j, y_j \mid x_i) = \log \frac{p(y_i \mid x_i, x_j, y_j)}{p(y_i \mid x_i)}.
> $$
>
> For the detailed proof and formal derivation, please refer to **Theorem 1** in the revised manuscript. This theoretical connection directly links our utility metric to an information-theoretic measure, demonstrating that \( UF_{ij} \) quantifies the informational gain from in-context examples.
>
> **For practical implementation and computational efficiency**, we adopt the Euclidean distance as the distance metric in place of KL-divergence:
>
> $$
> UF_{ij} = \left\| \mathbf{1} - p(y_i \mid x_i) \right\|_2 - \left\| \mathbf{1} - p(y_i \mid x_i, x_j, y_j) \right\|_2,
> $$
>
> where $\mathbf{1}$ is a vector of ones corresponding to the ground truth distribution $GT_i$, and $\left\| \cdot \right\|_2$ denotes the Euclidean norm.
>
> In-context learning (ICL) provides an efficient means to estimate these probabilities without incurring the computational costs associated with gradient-based methods. Despite using a simpler distance metric in practice, the utility metric retains its connection to information theory and serves as a robust measure of the impact of in-context learning on model behavior. Our experimental results across three diverse use cases empirically validate that the ICL-derived utility metric correlates with fine-tuning gains, confirming that the utility derived from ICL can be a reliable predictor of improvements in fine-tuning.
>
> ---
> **Q2: "utility gain reflecting sample diversity and model changes"**
>
> **A2**. Let us clarify two key aspects of DELIFT's utility-based selection:
>
> **Sample Diversity**: Rather than relying on semantic similarity, our utility kernel captures **information-based diversity** - selecting samples that provide complementary information for improving model performance. This is evidenced by DELIFT consistently outperforming DELIFT (SE), which uses only semantic embeddings, demonstrating the advantage of model-based diversity over purely semantic diversity.
>
> **Static vs. Dynamic Selection**: DELIFT currently performs **one-time subset selection** before fine-tuning, focusing on data efficiency through initial pruning. While the utility values of samples may change as the model is fine-tuned on the selected subset, adapting the selection during training introduces additional challenges: (1) recomputing utilities after each update is computationally expensive, (2) frequent changes to training data may impact convergence, and (3) maintaining consistent performance across training requires careful balance between exploration and exploitation. We leave the development of **efficient adaptive selection strategies** as future work, particularly for continual learning scenarios where the model's capabilities evolve significantly over time.

---

> ### Author Response · Authors · 2024-11-18
>
> **Q3: "analysis of diversity-based selection methods"**
>
> **A3.** To address the need for additional diversity-based comparisons, we included **DEFT-UCS** as a baseline, which uses sentence embeddings with centroid-based clustering to select diverse subsets. As shown in Tables 10 and 11, DELIFT significantly outperforms DEFT-UCS (LAJ scores: 2.98 vs 2.59), highlighting that our utility-based kernel is more effective at identifying informative samples than purely embedding-based clustering approaches.
>
> To better understand how different methods select examples for in-context learning, we analyze the top-2 examples chosen for the query "What are popular street foods from Maharashtra in India?. The experimental setting is the same as Table 11: (Llama-3.2-3B, Use Case 1, Mix-Instruct). Due to the space limitation, just the instruction/inputs are shown.
>
> Table 13: Top-2 ICL examples from each baseline given the input "What are popular street foods from Maharashtra in India?"
> ||Top-1st|Top-2nd|
> -|:-:|:-:|
> |Random|can you give me a few name for an auction house that would be more appeal to social media|Name five English irregular verbs.|
> |SelectIT|I'm trying to find a new restaurant. Can you recommend one?|Can you recommend any good books on the topic of meditation?|
> |DEFT-UCS|Give me a list of popular tourist attractions in Rome|Name five English irregular verbs.|
> |LESS|I'm a vegetarian. What are some of your favorite vegetarian recipes?|Give me a list of popular tourist attractions in Rome|
> |DELIFT (SE)|I'm trying to find a new restaurant. Can you recommend one?|Can you recommend any good books on the topic of meditation?|
> |DELIFT (FL Only) |I'm a vegetarian. What are some of your favorite vegetarian recipes?|Generate 15 4-letter words related to cooking.|
> |DELIFT|I'm trying to find a new restaurant. Can you recommend one?|I'm a vegetarian. What are some of your favorite vegetarian recipes?|
> |Full Data|I'm trying to find a new restaurant. Can you recommend one?|I'm a vegetarian. What are some of your favorite vegetarian recipes?|
>
> The analysis reveals several key patterns:
> 1. **Task Alignment**: Most methods except Random and DEFT-UCS select a recommendation-focused first example, but vary significantly in relevance:
>    - **DELIFT** maintains food-domain focus through restaurant recommendations and recipes
>    - **DEFT-UCS** drifts to general tourism and language tasks
>    - **DELIFT (SE)** and SelectIT stay with recommendations but lose food context in second example
>
> 2. **Complementary Information**: Methods differ in second example selection:
>    - **DELIFT** combines restaurant recommendation with vegetarian recipes, covering both general and specific food queries
>    - **DELIFT (FL)** maintains food context but with less recommendation focus
>    - Other methods either repeat recommendation structure (SelectIT) or select topically unrelated examples
>
> This analysis demonstrates how DELIFT's utility-based selection achieves better **domain relevance** and **information diversity** compared to both traditional clustering (DEFT-UCS) and embedding-based approaches. We will include this comprehensive comparison in the revised paper.
>
> ---
>
> **Q4: "analysis of different Facility Location variants"**
>
> **A4**. Thank you for this insightful suggestion. We conducted a thorough ablation study comparing vanilla Facility Location (FL) with our specialized variants across different fine-tuning stages. As shown in Table 11, for task-specific fine-tuning, **DELIFT with FLMI** (Facility Location Mutual Information) achieves a **2.61\% improvement** over **DELIFT with FL only** (LAJ scores: 2.71 vs 2.63). The qualitative analysis in Table 13 further illustrates this difference:
>
> - **FL Only**: Selects generally food-related examples ("vegetarian recipes", "cooking-related words")
> - **FLMI**: Selects more targeted examples combining recommendations and food context ("restaurant recommendation", "vegetarian recipes")
>
> This improvement validates our design choice of using specialized FL variants for different stages:
> - **FL**: Promotes general diversity
> - **FLMI**: Adds mutual information term to align with target task
> - **FLCG**: Incorporates conditional gain for continual learning
>
> We will expand the ablation analysis in our revision to demonstrate the benefits of each specialized variant across all fine-tuning stages.

---

> ### Author Response · Authors · 2024-11-18
>
> **Q5: "What is \( s_{ij} \)?"**
>
> **A5.** In our paper, the term \( s_{ij} \) can refer to two different kernels, depending on the context:
>
> 1. **Utility-based kernel (\( UF_{ij} \))**: This is the primary kernel used in our proposed DELIFT method. It quantifies the utility gain from selecting a sample, based on its contribution to improving the model’s performance.
>
> 2. **Pairwise similarity kernel**: This kernel is used in our baseline method, DELIFT (SE), as discussed in Section 4.2, point (4). It measures the semantic similarity between samples.
>
> The current version of DELIFT employs a **static subset selection** process, where uninformative data examples are pruned at the beginning of the fine-tuning stage. The goal of this approach is to maximize data efficiency by minimizing the number of different samples the model encounters during training. To achieve this, the utility-based kernel \( UF_{ij} \) is computed once, before fine-tuning begins.
>
> However, we acknowledge that a **dynamic subset selection** approach—where utility values and subsets are updated regularly throughout training—could potentially improve performance. This dynamic approach, however, would increase the number of samples the model processes during training, as well as the computational costs associated with subset selection. We view this as an interesting direction for future work, as it would allow the selection process to adapt to the evolving model, potentially improving its ability to learn from the most informative samples at each stage of fine-tuning.
>
> ---
>
>
> **Q6: "How is the ground truth distribution determined?"**
>
> **A6.** Thank you for the insightful question. The objective of our fine-tuning process is to align the model's predicted response with the ground truth. One effective way to achieve this is by matching the token distribution of the language model to that of the ground truth.
>
> In an ideal scenario, where the language model generates the exact correct response, the token distribution would assign a probability of 1 to the correct token and 0 to all other tokens in the vocabulary. Therefore, our "ground truth token distribution" is represented as a vector of 1’s, which reflects the desired probabilities for the correct token.
>
> We will provide further clarification on this in the revised version of our paper.
>
> ---
>
> **Q7: "The template is modified, lacks line numbers"**
>
> **A7.** Thank you for pointing that out. I inadvertently used an older template without line numbers. We have since updated the manuscript to include the latest template with line numbers. Apologies for any confusion or inconvenience caused.
>
> ---
>
> **Q8: "Why instruction-finetuned models instead of pre-trained models?"**
>
> **A8.** This is a valid concern. For more details, please refer to Tables 10 and 11 in our general comment, where we present experiments using Llama-3.2-3B and Mistral-7B-v0.1 models. The results from these experiments are consistent with those presented in the paper, highlighting the **effectiveness of DELIFT over all considered baselines**.
>
>
> ---
>
> We thank the reviewer for their time. Please let us know of any more questions and concerns you might have - we are happy to answer them. If we addressed your concerns sufficiently, we would appreciate to see this reflected in an increase of rating scores. Thank you again!

---

> > ### Comment · Reviewer_hvNC · 2024-11-22
> >
> > I thank the authors for their detailed reply. Most of my concerns are addressed and I will adjust my score accordingly.

---

> > > ### Author Response · Authors · 2024-11-22
> > >
> > > Thank you for your thoughtful feedback and for recognizing the improvements in our work. We strive for the highest quality and would greatly value your insights on any remaining areas we could enhance to fully meet your expectations.

---

### Official Review · Reviewer_gaDz · 2024-11-04

**Soundness:** 3
**Presentation:** 3
**Contribution:** 3
**Rating:** 6
**Confidence:** 4

**Summary:**

This paper presents DELIFT, Data Efficient Language model Instruction Fine-Tuning, a method to select and prune fine-tuning data for three stages - initial instruction tuning, task-specific fine-tuning, and continual fine-tuning. One of the contributions of the paper is a utility-based kernel to quantify the importance of a data point in the presence of another data point used as an in-context learning (ICL) sample. The authors further use various submodular functions (Facility Location - FL, Facility Location Mutual Information - FLMI, and Facility Location Conditional Gain - FLCG) for the three fine-tuning stages defined above. The main core of DELIFT comprises of a greedy algorithm to iteratively select data points that provide the maximum marginal gain with respect to the chosen submodular function. DELIFT shows clear gains over various baselines for all three stages of fine-tuning using the Phi-3 (3.8B params) and Qwen-2 (72B params).

**Strengths:**

- The paper proposes a novel utility metric function to measure mutual information of data points and their relevance. The authors further highlight the benefit of this choice by comparing against sentence embedding based metrics. DELIFT performs better compared to DELIFT used with SE.
- The authors build targeted modular functions for each stage (FL for standard IFT, FLMI for task-specific FT, and FLCG for continual FT), catering to the optimization objectives for these 3 stages.
- DELIFT is able to retain and sometimes improve performance over the full fine-tuning by pruning out 70% of the full data.

**Weaknesses:**

- Since the authors use post-trained models like Phi-3-mini-128k-instruct and Qwen2-72B-Instruct as the starting point, aren't the three stages described and the followup experiments basically a part of continual fine-tuning? I would be interested to see the effect of DELIFT when starting from base models and see if the method is able to prune out fine-tuning data for IFT starting from scratch.
- The authors present the results on ICL and QLoRA fine-tuning but not the full parameter fine-tuning. Are there diminishing gains with full model fine-tuning? I would like to see the results corresponding to those too.
- What's the computational cost for the data selection algorithm since it involves doing forward passes over the language model and not use cheap embedding models to compute the probability distributions. How does it compare it with the fine-tuning stage involving QLoRA.
- Weakness 1 is further enhanced when results for use case 2 are presented, where fine-tuning actually leads to drop in performance on MMLU, where it doesn't make sense to do any task-specific fine-tuning at all. And I don't trust the results on MT-Bench since it's a very small dataset and the error bars would be huge. To mitigate this weakness, I would suggest the authors to try task-specific fine-tuning on say some reasoning dataset and then track the performance on benchmarks like MATH/GSM8K/ARC-Challenge etc.

**Questions:**

Other than the questions listed in the weaknesses, here are few clarifying questions for the authors:

1. For computing the metric in equation 2, are the probability vectors ($d$ dimension) from all predicted tokens (say $x$ tokens) concatenated to form a vector of dimension $x * d$ and then the equation 2 is applied?
2. Minor typo in Figure 2 (should be LAJ scores instead of LJA scores).
3. Can some other baselines be also included like [1] and [2].


[1] DEFT-UCS: Data Efficient Fine-Tuning for Pre-Trained Language Models via Unsupervised Core-Set Selection, Das et al., 2024.

[2] Text Quality-Based Pruning for Efficient Training of Language Models, Sharma et al., 2024.

---

> ### Author Response · Authors · 2024-11-18
>
> Thank you for your feedback, and recognition of our novel utility metric and improvement over existing sentence embedding-based metrics.
>
> **Q1: "effect of DELIFT when starting from base models"**
>
> **A1**. This is a valid concern. Please refer to Tables 10 and 11 in our general comment. We used DELIFT on Llama-3.2-3B and Mistral-7B-v0.1 base models and find a similar trend in performance as in the paper with the instruction-tuned models. This showcases the effectiveness of DELIFT to select informative subsets even for base models that are not instruction fine-tuned.
>
> ---
>
> **Q2: "diminishing gains with full model fine-tuning?"**
>
> **A2**. Thank you for raising this important question. While our original experiments focused on QLoRA due to computational constraints, we have now conducted additional experiments with full fine-tuning using the Facebook opt-125m model (see Table 12). Our choice to primarily use QLoRA was motivated by three key considerations:
>
> 1. **Method Independence**: DELIFT's selected subsets demonstrate consistent effectiveness across a spectrum of use cases - from parameter-free approaches like in-context learning to parameter-efficient (QLoRA) and full model fine-tuning. This versatility highlights that our subset selection strategy captures inherently valuable training examples regardless of how they are ultimately used.
>
> 2. **Current Best Practices**: Parameter-efficient fine-tuning (PEFT) methods like QLoRA represent the current state-of-the-art approach for adapting large language models, offering a practical balance between computational efficiency and performance gains.
>
> 3. **Empirical Validation**: Our new results with opt-125m show that DELIFT consistently outperforms all baselines for both QLoRA and full fine-tuning settings. Notably, the performance gap between QLoRA and full fine-tuning is minimal, suggesting that in many cases, QLoRA may be the more practical choice given its significantly lower computational requirements.
>
> These findings demonstrate that DELIFT's benefits are robust across fine-tuning approaches while supporting our focus on QLoRA for large-scale experiments.
>
> Table 12: opt-125m on Use Case 1: Mix-Instruct (equivalent to Table 1).
>
> | Method      |       | QLoRA |      |       | Full FT |      |
> |-------------|-------|-------|------|-------|---------|------|
> |             | ROUGE | BGE   | LAJ  | ROUGE | BGE     | LAJ  |
> | Initial     | 9.04  | 40.50 | 1.19 | 9.57  |  40.86  | 1.14 |
> | Random      | 12.55 | 46.99 | 1.25 | 13.07 |  47.91  | 1.30 |
> | SelectIT    | 14.78 | 49.80 | 1.26 | 15.35 |  50.42  | 1.28 |
> | DEFT-UCS    | 15.12 | 50.29 | 1.39 | 15.16 |  50.29  | 1.39 |
> | LESS        | 15.52 | 50.70 | 1.38 | 16.02 |  51.30  | 1.40 |
> | DELIFT (SE) | 18.81 | 55.02 | 1.35 | 17.06 |  55.93  | 1.38 |
> | DELIFT      | 19.72 | 57.98 | 1.43 | 19.87 |  58.11  | 1.45 |
> | Full Data   | 20.39 | 60.39 | 1.95 | 21.64 |  61.70  | 2.05 |
>
> ---
>
> **Q3: "computational cost of data selection algorithm"**
>
> **A3**. Thank you for this important question about computational efficiency. While DELIFT's utility computation requires **O(n²) forward passes** through the language model (where **n** is the total number of training examples), this is a **one-time preprocessing cost** that can be amortized across multiple use cases - the **selected subsets** can be stored and reused for different fine-tuning approaches (ICL, QLoRA, full fine-tuning), future model updates, and continuous dataset refinements. The actual subset selection process using submodular optimization is **highly efficient**, requiring only **O(kn) computations** (where **k** is the desired subset size) with pre-computed utility scores. In practice, the total computational cost of DELIFT (utility computation and subset selection) takes about **50\% of the time needed for QLoRA fine-tuning on the full dataset**, making it particularly cost-effective when considering full fine-tuning (which requires significantly more compute than QLoRA) or when maintaining and updating training sets over time. This makes DELIFT a **valuable tool** for teams looking to maintain high-quality, reusable training sets while optimizing their computational resources.

---

> ### Author Response · Authors · 2024-11-18
>
> **Q4: "task-specific fine-tuning on reasoning datasets"**
>
> **A4**. Thank you for this valuable feedback about evaluation methodology. We acknowledge your concerns about MT-Bench's small size and the MMLU performance drop. To address this, we conducted extensive experiments with **GSM-8k**, a comprehensive mathematical reasoning benchmark. As shown in Table 11, DELIFT demonstrates strong performance - achieving **2.71 LAJ score** with only 30\% of the data, compared to 2.85 for full data, while consistently **outperforming all baselines** across both ICL and QLoRA settings. Importantly, even when starting from a relatively low baseline performance (LAJ: 1.36), DELIFT-selected subsets enable significant improvements, validating our approach for task-specific fine-tuning on structured reasoning tasks. We will expand our evaluation to include additional reasoning benchmarks like MATH Hard and ARC-Challenge in the final version.
>
> ---
>
> **Q5: "probability vectors from all predicted tokens?"**
>
> **A5**. Thank you for this clarifying question. For each token position in the sequence of length **x**, we compute the model's probability distribution over the entire vocabulary. Then, for each position, we extract only the probability assigned to the ground truth token, resulting in an **x-dimensional vector**. Using teacher forcing during generation ensures accurate probability estimation at each step. This approach is computationally efficient as it avoids working with full vocabulary-sized distributions while capturing how well the model predicts the correct tokens in sequence. The distance metric in equation 2 then compares this x-dimensional probability vector against a **vector of ones** (representing perfect prediction where each ground truth token should have probability 1).
>
> ---
>
> **Q6: "typo in Figure 2"**
>
> **A6**. Thank you for catching this error. We have corrected "LJA scores" to "LAJ scores" (LLM-as-Judge) in Figure 2 and verified all other instances in the paper for consistency.
>
> ---
>
> **Q7: "additional baselines comparison"**
>
> **A7**. Thank you for suggesting these relevant works. We have incorporated **DEFT-UCS** as a baseline in our experiments - see Tables 10 and 11. While DEFT-UCS uses sentence embeddings to select diverse subsets through centroid-based clustering, our results show that **DELIFT significantly outperforms it** (LAJ scores: 2.98 vs 2.59 in Table 10). This highlights that DELIFT's utility-based kernel is more effective at identifying informative samples than purely embedding-based approaches. Regarding the Text Quality paper, while it focuses on filtering grammatically incorrect samples for pre-training, our work addresses a fundamentally different aspect of data quality - **selecting informative examples that improve model performance during fine-tuning**. While grammatical correctness is important, DELIFT optimizes for examples that actively enhance the model's capabilities. We will include a detailed comparison with both approaches in our related work section.
>
> ---
>
> We thank the reviewer for their time. Please let us know of any more questions and concerns you might have - we are happy to answer them. If we addressed your concerns sufficiently, we would appreciate to see this reflected in an increase of rating scores. Thank you again!

---

> ### Author Response · Authors · 2024-11-22
>
> Thank you for taking the time to review our work. We note that your feedback aligns with some of the concerns raised by other reviewers, which we have addressed through additional experiments detailed in our revised manuscript and initial response. If you have any further questions or concerns, we would be happy to address them. If our revisions sufficiently address your feedback, we would appreciate your consideration of this in your evaluation. Thank you again for your efforts in reviewing our work!

---

> ### Comment · Reviewer_gaDz · 2024-11-27
> **Thank you for additional results**
>
> Given my own experiences with supervised fine-tuning, I would always prefer full fine-tuning compared to data-efficient fine-tuning techniques, because you have to consider performance across all benchmarks at once and not look at just each benchmark separately. However, given hard resource constraints, the paper's methods, results, and additional results during the rebuttal stage have convinced me of the effectiveness of their approach, and therefore, I am raising my score to a 6.

---

> > ### Author Response · Authors · 2024-11-27
> >
> > Thank you for your consideration and for recognizing our reasoning behind parameter-efficient fine tuning. We strive for the highest quality and would greatly value your insights on any remaining areas we could enhance to fully meet your expectations.

---

### Public Comment · ~Junyu_Luo2 · 2024-11-13
**Nice Work of Data-efficient LLM**

I greatly enjoyed reading the authors' impressive work. The research on improving LLM performance in data-efficient settings is both timely and inspiring. Given our shared research interests, I would like to humbly share our recent exploration in this direction[1]. We find our work is relevant to this research and would be deeply grateful for any discussion of our work in the future revisions.

Congratulations on the authors' excellent contribution to the field.

Best regards,

Junyu

---
[1]  SemiEvol: Semi-supervised Fine-tuning for LLM Adaptation.

---

> ### Author Response · Authors · 2024-11-21
>
> Thank you Junyu for taking the time to read our work, and the pointer to your paper. Your two-step methodology of using the labeled data for ICL/PEFT and using the unlabeled data for selecting confident responses from an ensemble of models for PEFT is interesting!

---

### Author Response · Authors · 2024-11-18
**Experiments with base models**

We thank the reviewers for their time and effort in reviewing our work. We especially appreciate the fact that the reviewers found our work to be novel while (1) targeting specific use cases with a general framework, and (2) showcasing good results with DELIFT's subset selection methodology. We wish to address some common concerns that reviewers posed.

---
Below are the results of various subset selection baselines on Use Case 1: Instruction Tuning with Mix-Instruct as the dataset, and is equivalent to the setting in Table 1 of the paper. This is performed on the base model Llama-3.2-3B. As per reviewer gaDz's suggestion, we also add DEFT-UCS as a baseline. We see that **the takeaways are consistent with those mentioned in the paper**. Similar to DELIFT (SE), DEFT-UCS relies on sentence embeddings for sampling representative subsets. One key issue with DEFT-UCS is that it doesn't select representative samples useful for the model but rather rely on the semantics of the data samples. Similarly, our results show that **DELIFT (SE) outperforms DEFT-UCS** even though both rely on sentence embeddings for clustering highlighting the effectiveness of facility location as the subset selection objective. Finally, **DELIFT outperforms all the baselines**, highlighting the effectiveness of our utility kernel in capturing the important samples for the model.

Table 10: Llama-3.2-3B (base) on Use Case 1: Mix-Instruct (equivalent to Table 1).
| Method      |       | ICL   |      |       | QLoRA |      |
|-------------|-------|-------|------|-------|-------|------|
|             | ROUGE | BGE   | LAJ  | ROUGE | BGE   | LAJ  |
| Initial     | 25.88 | 61.97 | 1.9  | 28.51 | 73.14 | 2.48 |
| Random      | 28.64 | 77.14 | 2.59 | 38.79 | 79.89 | 2.53 |
| SelectIT    | 42.67 | 81.54 | 2.67 | 45.87 | 84.67 | 2.61 |
| DEFT-UCS    | 41.55 | 80.12 | 2.63 | 41.03 | 81.86 | 2.59 |
| LESS        | 44.99 | 82.48 | 2.69 | 50.54 | 84.14 | 2.78 |
| DELIFT (SE) | 51.19 | 83.54 | 2.72 | 55.32 | 85.92 | 2.79 |
| DELIFT      | 54.58 | 88.29 | 2.83 | 58.57 | 90.98 | 2.98 |
| Full Data   | 55.46 | 92.71 | 3.31 | 61.23 | 95.52 | 3.10 |

---
Below are the results of various subset selection baselines on Use Case 2: Task-Specific Fine-Tuning where examples from Mix-Instruct that are similar to those in GSM-8k are chosen in the subsets. This is equivalent to Table 4 in our paper, and is performed on the base model Mistral-7B-v0.1. As with Table 10, **the takeaways here are also consistent with those from the paper**. As per reviewer hvNC's suggestion, we conducted an ablation study comparing two variants of our method: **DELIFT with Facility Location Mutual Information (FLMI)** and **DELIFT with only Facility Location (FL)**. The results demonstrate that **DELIFT with FLMI achieves superior performance**, yielding a 2.61\% improvement over the FL-only variant, validating our choice of FLMI for task-specific fine-tuning. Notably, even **DELIFT with FL alone outperforms all other baselines**, underscoring the fundamental strength of our utility kernel and facility location objective for subset selection.

Table 11: Mistral-7B-v0.1 (base) on Use Case 2: Mix-Instruct and GSM-8k (equivalent to Table 4).
| Method           |       | ICL   |      |       | QLoRA |      |
|------------------|-------|-------|------|-------|-------|------|
|                  | ROUGE | BGE   | LAJ  | ROUGE | BGE   | LAJ  |
| Initial          | 17.06 | 55.13 | 1.35 | 27.95 | 65.92 | 1.36 |
| Random           | 19.18 | 56.03 | 1.38 | 33.85 | 81.74 | 1.85 |
| SelectIT         | 30.27 | 77.23 | 1.45 | 42.29 | 88.17 | 2.49 |
| DEFT-UCS         | 30.06 | 76.99 | 1.59 | 41.45 | 87.84 | 2.08 |
| LESS             | 31.69 | 77.87 | 2.26 | 43.86 | 88.22 | 2.60 |
| DELIFT (SE)      | 31.84 | 78.62 | 2.44 | 43.04 | 90.53 | 2.54 |
| DELIFT (FL Only) | 32.30 | 78.54 | 2.54 | 44.62 | 91.04 | 2.63 |
| DELIFT           | 33.25 | 79.10 | 2.56 | 46.12 | 92.97 | 2.71 |
| Full Data        | 35.33 | 81.59 | 2.79 | 49.10 | 94.57 | 2.85 |

---

### Meta-Review · Area_Chair_YmJT · 2024-12-24

**Metareview:**

## Summary:
The paper introduces DELIFT (Data Efficient Language model Instruction Fine-Tuning), an algorithm designed to optimize data selection efficiently across three crucial stages of fine-tuning large language models (LLMs). DELIFT utilizes a pairwise utility metric, which uses in-context learning to quantify the significance of each data sample in enhancing the model's responses to other samples, enabling informed data selection. By applying different facility-location-based submodular functions tailored to each fine-tuning stage, DELIFT strategically selects diverse and optimal subsets, reducing fine-tuning data requirements by up to 70% without compromising performance. Experiments across various tasks and model scales validate DELIFT's computational efficiency and superior efficacy compared to existing methods, demonstrating clear gains in both efficiency and effectiveness throughout the fine-tuning process.

## Strengths:
1. DELIFT studies the important data selection problem for all three fine-tuning stages of LLMs using submodular maximization of facility location functions, which is a novel application of submodular optimization.
1. The in-context learning-based utility metric is novel: it measures the contribution of sample-j to predicting sample-i in order to quantify the mutual information and relevance between the two samples.
1. It introduces three facility location functions to measure the utility of selected subsets of data for the three finetuning stages, respectively, by providing intuitive explanations.
1. DELIFT demonstrates the ability to reduce fine-tuning data requirements by up to 70% while maintaining or even improving performance.

## Weaknesses:
1. The effectiveness of the proposed method needs to be demonstrated in more general cases, e.g., finetuning base models, full-model finetuning, and comparisons to more recent data selection baselines. The rebuttal provides some relevant and new experiments.
1. It lacks a thorough comparison of the computational cost for data selection. Most existing data selection approaches for LLM finetuning only require $O(n)$ forward inferences for $n$ samples and evaluating $n$ scores followed by top-$k$ selection. But the proposed method requires $O(n^2)$ forward inferences to estimate the pairwise utility and $O(kn)$ evaluations for selecting $k$ samples. Hence, the proposed method can be much more expensive. This is the diversity tax that may undermine the practical applicability of the method.
1. Although the goal is ambitious and targets three finetuning stages, the comparison for each stage does not cover some widely-used datasets, e.g., Alpaca/UltraChat/MAGPIE for instruction tuning, FLAN/SuperGLUE for task-specific finetuning (MMLU and MT-Bench are evaluation benchmarks with a small data size per task). The presented GSM-8k is better but a thorough report is needed.
1. The choice of submodular functions and hyperparameters is usually highly important to their practical performance. The paper only focuses on facility location-based functions but lacks discussion and comparison to other widely studied ones in submodular optimization literature.
1. The paper does not discuss the relationship and difference between the proposed ICL-based utility in Eq. 1 and the influence function (Koh & Liang, ICML'17), both aiming to measure the gain of one sample bringing to another in training.

## Decision:
Two reviewers raised their ratings from 5 to 6 after the rebuttal and confirmed some important concerns in their original comments have been successfully addressed by the rebuttal. The authors are encouraged to provide more thorough and complete evaluations on more widely used benchmarks and keep improving the paper based on the discussion. Based on the positive feedback of all reviewers, the meta-reviewer recommends acceptance of the paper.

**Additional Comments On Reviewer Discussion:**

In the original comments, the reviewers raised several concerns mainly in experimental settings, baselines, models to finetune, ablation study on FL, etc. They also find the discussion of the ICL and diversity insufficient. The authors provided further clarifications and additional experimental results in the rebuttal, which addressed several major concerns. After the rebuttal, two reviewers out of the three raised their ratings from 5 to 6. This reflects the effectiveness of the responses by the authors.

---

### Decision · Program_Chairs · 2025-01-22

Accept (Poster)